# A full Stokes subgrid scheme in two dimensions for simulation of grounding line migration in ice sheets using Elmer/ICE (v8.3)

Gong Cheng[1], Per Lötstedt[1], and Lina von Sydow[1]

[1]Department of Information Technology, Uppsala University, P. O. Box 337, SE-75105 Uppsala, Sweden

**Correspondence:** Gong Cheng (cheng.gong@it.uu.se)

**Abstract.** The flow of large ice sheets and glaciers can be simulated by solving the full Stokes equations using a finite element method. The simulation is particularly sensitive to the discretization of the grounding line, which separates ice resting on bedrock and ice floating on water, and is moving in time. The boundary conditions at the ice base are enforced by Nitsche's method and a subgrid treatment of the grounding line element. Simulations with the method in two dimensions for an advancing and a retreating grounding line illustrate the performance of the method. The computed grounding line position is compared to previously published data with a fine mesh, showing that similar accuracy is obtained using subgrid modeling with more than 20 times coarser meshes. This subgrid scheme is implemented in the two dimensional version of the open source code Elmer/ICE.

## 1 Introduction

### 1.1 Ice sheet dynamics, sea-level rise, and grounding line migration

Numerical simulation of ice sheet flow is necessary to assess the future sea-level rise (SLR) due to melting of continental ice sheets and glaciers (Hanna et al., 2013) and to reconstruct the ice sheets of the past (Stokes et al., 2015; DeConto and Pollard, 2016) for comparison with measurements and validation of the models. Ice sheet model predictions are particularly sensitive to the numerical treatment of the grounding line (GL) (Durand and Pattyn, 2015; Konrad et al., 2018), the line where the ice sheet leaves the solid bedrock and becomes an ice shelf floating on water driven by buoyancy.

The distance that the GL moves may be long over palaeo time scales. In Kingslake et al. (2018) it is shown that the GL has retreated several hundred kilometers in West Antarctica during the last 11,500 years and then advanced again after the isostatic rebound of the bed. The sensitivity, long time intervals, and long distances of the GL migration require a careful treatment of the GL and its neighborhood in the numerical method used to discretize the equations modeling the ice sheet dynamics. In this paper, we develop an accurate and efficient method for such problems.

### 1.2 Model equations

When the ice rests on the ground and is affected by large frictional forces on the bed, the ice flow is dominated by vertical shear stresses. On the other hand, when the ice is floating on water, it is the longitudinal stress gradient that controls the flow

of the ice. The GL is in the transition zone between these two types of flow with a gradual change of the stress field (Schoof, 2011).

The most accurate ice model in theory is based on the full Stokes (FS) equations. A simplification of the FS equations by integrating in the depth of the ice is the shallow shelf (or shelfy stream) approximation (SSA) (MacAyeal, 1989), which is often used for simulation of the coupling between a grounded ice sheet and a marine ice shelf. In the zone between the grounded ice and the floating ice, it is necessary to use the FS equations (Wilchinsky and Chugunov, 2000; Schoof and Hindmarsh, 2010; Docquier et al., 2011; Schoof, 2011) unless the ice is moving rapidly on the ground with low basal friction, when the SSA equations are accurate both upstream and downstream of the GL.

The evolution of the GL in simulations is sensitive to the model equations and the basal friction law. In the Marine Ice Sheet Model Intercomparison Project (MISMIP) (Pattyn et al., 2012, 2013), different ice models and implementations solve the same ice flow problems and the predicted GL steady state and transient GL motion are compared. The results show that the position of the GL depends on the model equations (Pattyn et al., 2013). Predictions of the GL position and SLR is different for different ice models such as FS and SSA (Pattyn and Durand, 2013). Including equations with vertical shear stress at the GL such as the FS equations is crucial to accurately resolve GL dynamics in a wide range of circumstances.

The friction laws at the ice base depend on the effective pressure, the basal velocity, and the distance to the GL in different combinations in Leguy et al. (2014); Gagliardini et al. (2015); Brondex et al. (2017); Gladstone et al. (2017). The GL position and SLR vary considerably depending on the choice of friction law. Given the friction law, the results are sensitive to its model parameters too (Gong et al., 2017).

### 1.3 Numerical methods

Parameters in the numerical methods used to simulate ice sheet flow influence the GL migration. Durand et al. (2009b) find that the mesh resolution along the ice bed has to be fine to obtain reliable solutions with FS in GL simulations. The GL is then located in a node of the fixed or static mesh. A mesh size below 1 km is necessary in Larour et al. (2019) to resolve the features at the GL. Adaptive meshes for a finite volume discretization of an approximation of the FS equations are employed in Cornford et al. (2013) to study the GL retreat and loss of ice in West Antarctica. The FS solutions of benchmark problems in Pattyn et al. (2013) computed by an implementation of the finite element method (FEM) in Elmer/ICE (Gagliardini et al., 2013) and FELIX-S (Leng et al., 2012) are compared in Zhang et al. (2017). The differences between the these implementations are attributed to different treatment of a friction parameter at the GL and different assignment of grounded and floating nodes and element faces.

A subgrid scheme introduces an inner structure in the discretization element or mesh volume where the GL is located. Such schemes have been developed for simplifications of the FS equations. A subgrid model for the GL is tested in Gladstone et al. (2010b) for the one dimensional (1D) SSA equation where the flotation condition for the ice defines the position of the GL. The GL migration is determined by the two dimensional (2D) SSA equations discretized by the finite element method (FEM) in Seroussi et al. (2014). Subgrid models at the GL are compared to a model without an internal structure in the element. The conclusion is that sub-element parameterization is necessary to obtain accurate results for reasonable computational expense.

A shallow approximation to FS with a subgrid scheme on coarse meshes is compared to FS in Feldmann et al. (2014) with similar results for the GL migration. Subgrid modeling and adaptivity are compared in Cornford et al. (2016) for a vertically integrated model. The thickness of the ice above flotation determines if the ice is grounded or floating. A fine mesh resolution is necessary for converged GL positions with FS in Durand et al. (2009a, b). A dynamic mesh refinement and coarsening of the mesh following the GL would solve the problem in palaeo simulations when the GL moves long distances. An alternative is to introduce a subgrid scheme in the mesh elements where the GL is located in a static mesh and keep the mesh size coarser everywhere else in the ice sheet.

## 1.4 Proposed method and outline of the paper

From the above we conclude that, it seems crucial that the ice model includes equations with vertical shear stress in the neighbourhood of the GL, and one way to avoid the fine meshes that are otherwise needed close to the GL is to introduce a subgrid scheme in the discretization element where the GL is located. In this study, we develop such a numerical method for the FS equations in two dimensions introducing a subgrid scheme in the mesh element where the GL is located. Since the subgrid scheme is restricted to one element in a 2D vertical ice this is computationally inexpensive. In an extension to 3D, the subgrid scheme would be applied along a line of elements in 3D. The results with numerical modeling will always depend on the mesh resolution but can be more or less sensitive to the mesh spacing and time steps. It depends on the equation, the mesh size, the mesh quality, and the finite element spaces in the approximation.

We solve the FS equations in a 2D vertical ice with the Galerkin method implemented in Elmer/ICE (Gagliardini et al., 2013). A subgrid discretization is proposed and tested for the element where the GL is located. The boundary conditions are imposed by Nitsche's method at the ice base in the weak formulation of the equations (Nitsche, 1971; Urquiza et al., 2014; Reusken et al., 2017). The linear Stokes equations are solved in Chouly et al. (2017a) with Nitsche's treatment of the boundary conditions. They solve the equations for the displacement but here we solve for the velocity using similar numerical techniques to weakly impose the Dirichlet boundary conditions on the normal velocity at the base. The frictional force in the tangential direction is applied on part of the element with the GL. The position of the GL within the element is determined in agreement with theory developed for the linearized FS in Schoof (2011).

The paper is organized as follows. Section 2 is devoted to the presentation of the mathematical model of the ice sheet dynamics. In Sect. 3, the numerical discretization with FEM is given while the subgrid scheme around the GL is found in Sect. 4. The numerical results for a MISMIP problem are presented in Sect. 5. The extension to three dimensions (3D) is discussed in Sect. 6 and finally some conclusions are drawn in Sect. 7.

## 2    Ice model

### 2.1    The full Stokes (FS) equations

To simulate flow in a 2D vertical cross-section of an ice sheet, we use the FS equations with coordinates $\boldsymbol{x} = (x,z)^T$ (Hutter, 1983). The nonlinear partial differential equations (PDEs) in the interior of the ice domain $\Omega$ are given by

$$\begin{cases} \nabla \cdot \boldsymbol{u} = 0, \\ -\nabla \cdot \sigma = \rho \mathbf{g}, \end{cases} \tag{1}$$

where the stress tensor is $\sigma = \tau(\boldsymbol{u}) - p\mathbb{I}$ and the deviatoric stress tensor is $\tau(\boldsymbol{u}) = 2\eta(\boldsymbol{u})\dot{\epsilon}(\boldsymbol{u})$. The strain rate tensor is defined by

$$\dot{\epsilon}(\boldsymbol{u}) = \frac{1}{2}(\nabla \boldsymbol{u} + \nabla \boldsymbol{u}^T) = \begin{pmatrix} \dot{\epsilon}_{11} & \dot{\epsilon}_{12} \\ \dot{\epsilon}_{12} & \dot{\epsilon}_{22} \end{pmatrix}, \tag{2}$$

$\mathbb{I}$ is the identity matrix, and the viscosity is defined by Glen's flow law

$$\eta(\boldsymbol{u}) = \frac{1}{2}\left(\mathcal{A}(T')\right)^{-\frac{1}{n}} \dot{\epsilon}_e^{\frac{1-n}{n}}, \qquad \dot{\epsilon}_e = \sqrt{\frac{1}{2}\mathrm{tr}(\dot{\epsilon}(\boldsymbol{u})\dot{\epsilon}(\boldsymbol{u}))}. \tag{3}$$

Here $\boldsymbol{u} = (u,w)^T$ is the vector of velocities, $\rho$ is the density of the ice, $p$ denotes the pressure, and the gravitational vector is denoted by $\mathbf{g}$. The viscosity $\eta$ is a function of the rate factor $\mathcal{A}(T')$ where $T'$ is the ice temperature. For isothermal flow assumed here, the rate factor $\mathcal{A}$ is constant. Finally, $n$ is usually taken to be 3.

### 2.2    Boundary conditions

At the boundary $\Gamma$ of the ice domain $\Omega$ we define the normal outgoing vector $\boldsymbol{n}$ and tangential vector $\boldsymbol{t}$ (see Fig. 1). In the 2D vertical case considered here, the ice sheet geometry is constant in $y$. The ice surface is denoted by $\Gamma_s$ and the ice base is $\Gamma_b = \Gamma_{bg} \cup \Gamma_{bf}$. At $\Gamma_s$ and $\Gamma_{bf}$, the floating part of $\Gamma_b$, we have that

$$\sigma\boldsymbol{n} = \mathbf{f}_s \quad , \quad \sigma\boldsymbol{n} = \mathbf{f}_{bf}, \tag{4}$$

respectively. The ice is stress-free at $\Gamma_s$, $\mathbf{f}_s = 0$, and $\mathbf{f}_{bf} = -p_w\boldsymbol{n}$ at the ice/ocean interface $\Gamma_{bf}$ where $p_w$ is the water pressure. Let

$$\sigma_{\boldsymbol{nt}} = \boldsymbol{t} \cdot \sigma\boldsymbol{n}, \ \sigma_{\boldsymbol{nn}} = \boldsymbol{n} \cdot \sigma\boldsymbol{n}, \ u_{\boldsymbol{t}} = \boldsymbol{t} \cdot \boldsymbol{u},$$

where $\boldsymbol{\sigma_{nn}}$ and $\boldsymbol{\sigma_{nt}}$ are the normal and tangential components of the stress and $u_{\boldsymbol{t}}$ is the tangential component of the ice velocity at the ice base. Then for the slip boundary $\Gamma_{bg}$, the grounded part of $\Gamma_b$ where the ice rests on the bedrock, we have a

110 friction law for the sliding ice

$$\sigma_{\boldsymbol{nt}} + \beta(\boldsymbol{u},\boldsymbol{x})u_{\boldsymbol{t}} = 0, \quad u_{\boldsymbol{n}} = \boldsymbol{n} \cdot \boldsymbol{u} = 0, \quad -\sigma_{\boldsymbol{nn}} \geq p_w, \tag{5}$$

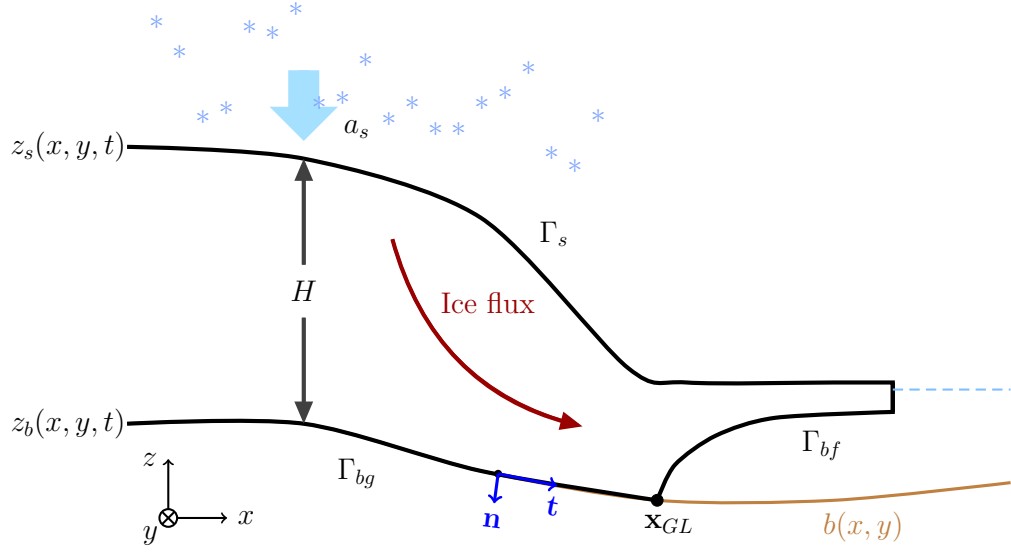

**Figure 1.** A two dimensional schematic view of a marine ice sheet.

where $u_n$ is the normal component of the ice velocity. The type of friction law is determined by the friction coefficient $\beta\ (\geq 0)$. At $\Gamma_{bf}$, there is a balance between $\sigma_{nn}$ and $p_w$ and the contact is friction-free, $\beta = 0$. Then

$$\sigma_{nt} = 0, \qquad -\sigma_{nn} = p_w. \tag{6}$$

At the GL, the boundary condition switches from $\beta > 0$ and $u_n = 0$ on $\Gamma_{bg}$ to $\beta = 0$ and a free $u_n$ on $\Gamma_{bf}$. In a 2D vertical cross-section of ice, the GL is the point $(x_{GL}, z_{GL})$ shared between $\Gamma_{bg}$ and $\Gamma_{bf}$.

    The ocean surface is at $z = 0$, and $p_w = -\rho_w g z_b$. The density of sea water is denoted by $\rho_w$, $z_b$ is the $z$-coordinate of $\Gamma_b$, and $g$ is the vertical component of the gravitational force.

## 2.3   The free surface equations

The boundaries $\Gamma_s$ and $\Gamma_b$ are time-dependent and move according to two free surface equations. The boundary $\Gamma_{bg}$ follows the fixed bedrock with coordinates $(x, b(x))$.

    The $z$-coordinate of the ice surface position $z_s(x, t)$ at $\Gamma_s$ (see Fig. 1) is the solution of an advection equation

$$\frac{\partial z_s}{\partial t} + u_s \frac{\partial z_s}{\partial x} - w_s = a_s, \tag{7}$$

where $a_s$ denotes the surface mass balance and $\boldsymbol{u}_s = (u_s, w_s)^T$ the velocity at the ice surface in contact with the atmosphere.

Similarly, the $z$-coordinate for the ice base $z_b$ of the floating ice at $\Gamma_{bf}$ satisfies

$$\frac{\partial z_b}{\partial t} + u_b \frac{\partial z_b}{\partial x} - w_b = a_b, \tag{8}$$

where $a_b$ is the basal mass balance and $\boldsymbol{u}_b = (u_b, w_b)^T$ the velocity of the ice at $\Gamma_{bf}$. On $\Gamma_{bg}$, $z_b = b(x)$ and on $\Gamma_{bf}$, $z_b > b(x)$.

    The thickness of the ice is denoted by $H = z_s - z_b$ and depends on $x$ and $t$.

## 2.4 A first order solution close to the grounding line

The 2D vertical solution of the FS equations in Eq. (1) with a constant viscosity, $n = 1$ in Eq. (3), is expanded in small parameters in Schoof (2011). The solutions in different regions around the GL are connected by matched asymptotics. Upstream of the GL at the grounded part, $x < x_{GL}$, the leading terms in the expansion satisfy a simple relation in scaled variables close to the GL. Across the GL, the ice velocity $u$, the flux of ice $uH$, and the depth-integrated normal or longitudinal stress $\tau_{11}$ in Eq. (2) are continuous. By including higher order terms in the expansion in small parameters, it is shown in Schoof (2011, Sect. 4.7) that the ice surface slope is continuous and Archimedes' flotation condition

$$H\rho = -z_b \rho_w \tag{9}$$

is not satisfied immediately downstream of the GL. A rapid variation in the vertical velocity $w$ in a short distance interval at the GL causes oscillations in the ice surface in the analysis as also observed in FS simulations in Durand et al. (2009a). The flotation condition in (9) determines where the GL is in SSA in Docquier et al. (2011); Drouet et al. (2013).

In Schoof (2011, Sect. 4.3), the solution to the FS in a 2D vertical cross-section of ice is expanded in two parameters, $\nu$ and $\epsilon$. The aspect ratio of the ice $\nu$ is the quotient between a typical scale of the thickness of the ice $\mathcal{H}$ and a horizontal length scale $\mathcal{L}$, $\nu = \mathcal{H}/\mathcal{L}$, and $\epsilon$ is $\nu$ times the quotient between the longitudinal and the shear stresses $\tau_{11}$ and $\tau_{12}$ in Eq. (2). If $\nu^{5/2} \ll \epsilon \ll 1$ then in a boundary layer close to the GL and $x < x_{GL}$ it follows from the equations that the leading terms in the solution in scaled variables satisfy

$$\tau_{22} - p = \sigma_{22} = \rho g(z - z_s). \tag{10}$$

On floating ice $\tau_{22} - p + p_w = 0$ and the hydrostatic flotation criterion Eq. (9) is fulfilled. This is a first order approximation of the second relation in Eq. (6). On the grounded ice domain, we have $\tau_{22} - p + p_w < 0$.

Introducing the notation

$$\chi_a(x, z) = \tau_{22} - p + p_w = \rho g(z - z_s(x)) - \rho_w g z_b(x), \tag{11}$$

and letting $H_{bw} = -z_b$ be the thickness of the ice below the sea level yields

$$\chi_a(x, z_b) = -g(\rho H - \rho_w H_{bw}). \tag{12}$$

If $x < x_{GL}$ then $\chi_a < 0$ in the neighborhood of $x_{GL}$ on $\Gamma_{bg}$ and if $x > x_{GL}$ then $\chi_a = 0$ and Eq. (9) holds true on $\Gamma_{bf}$. Suppose that $z_s$ and $z_b$ are linear in $x$. Then $\chi_a$ is also linear in $x$. In numerical experiments with the linear FS ($n = 1$) in Nowicki and Wingham (2008), $\chi_a(x, z_b)$ varies linearly in $x$ for $x < x_{GL}$.

In Sect. 4, we take this same approach but use an indicator $\chi(x)$ or $\tilde{\chi}(x)$ derived from the solutions of the nonlinear FS equations to estimate the GL position. These indicators are approximated by $\chi_a(x, z_b)$.

## 3 Discretization by FEM

In this section we state the weak form of Eq. (1), introduce the spatial FEM discretization used for Eq. (1), and give the time-discretization of Eq. (7) and (8).

 ### 3.1 The weak form of the FS equations

We start by defining the mixed weak form of the FS equations. Introduce $k = 1 + 1/n$, $k^* = 1 + n$ with $n$ from Glen's flow law and the spaces

$$\boldsymbol{V}_k = \{\boldsymbol{v} : \boldsymbol{v} \in (W^{1,k}(\Omega))^2\}, \quad Q_{k^*} = \{q : q \in L^{k^*}(\Omega)\}, \tag{13}$$

see, e.g. Jouvet and Rappaz (2011, Eq. (3.7)), Chen et al. (2013, Sect. 3.1), Martin and Monnier (2014, Eq. (21)). The weak solution $(\boldsymbol{u}, p)$ of Eq. (1) is obtained as follows. Find $(\boldsymbol{u}, p) \in \boldsymbol{V}_k \times Q_{k^*}$ such that for all $(\boldsymbol{v}, q) \in \boldsymbol{V}_k \times Q_{k^*}$ the equation

$$A((\boldsymbol{u}, p), (\boldsymbol{v}, q)) + B_\Gamma(\boldsymbol{u}, \boldsymbol{v}, p) + B_\mathcal{N}(\boldsymbol{u}, \boldsymbol{v}, q) = F(\boldsymbol{v}) + F_\Gamma(\boldsymbol{v}), \tag{14}$$

is satisfied, where

$$A((\boldsymbol{u}, p), (\boldsymbol{v}, q)) = \int_\Omega 2\eta(\boldsymbol{u})\dot{\epsilon}(\boldsymbol{u}) : \dot{\epsilon}(\boldsymbol{v}) \ \mathrm{d}\boldsymbol{x} - b(\boldsymbol{u}, q) - b(\boldsymbol{v}, p),$$

$$b(\boldsymbol{u}, q) = \int_\Omega q\nabla\cdot\boldsymbol{u} \ \mathrm{d}\boldsymbol{x},$$

$$B_\Gamma(\boldsymbol{u}, \boldsymbol{v}, p) = -\int_{\Gamma_{bg}} (\sigma_{\boldsymbol{nn}}(\boldsymbol{u}, p)\boldsymbol{n}\cdot\boldsymbol{v} + \sigma_{\boldsymbol{nt}}(\boldsymbol{u}, p)\boldsymbol{t}\cdot\boldsymbol{v}) \ \mathrm{d}s = \int_{\Gamma_{bg}} (-\sigma_{\boldsymbol{nn}}(\boldsymbol{u}, p)\boldsymbol{n}\cdot\boldsymbol{v} + \beta(\boldsymbol{t}\cdot\boldsymbol{u})(\boldsymbol{t}\cdot\boldsymbol{v})) \ \mathrm{d}s,$$

$$B_\mathcal{N}(\boldsymbol{u}, \boldsymbol{v}, q) = -\int_{\Gamma_{bg}} \sigma_{\boldsymbol{nn}}(\boldsymbol{v}, q)\boldsymbol{n}\cdot\boldsymbol{u} \ \mathrm{d}s + \gamma_0 \int_{\Gamma_{bg}} \frac{1}{h}(\boldsymbol{n}\cdot\boldsymbol{u})(\boldsymbol{n}\cdot\boldsymbol{v}) \ \mathrm{d}s,$$

$$F(\boldsymbol{v}) = \int_\Omega \rho\boldsymbol{g}\cdot\boldsymbol{v} \ \mathrm{d}\boldsymbol{x},$$

$$F_\Gamma(\boldsymbol{v}) = -\int_{\Gamma_{bf}} p_w\boldsymbol{n}\cdot\boldsymbol{v} \ \mathrm{d}s$$

The last term in $B_\mathcal{N}$ is added in the weak form in Nitsche's method (Nitsche, 1971) to impose the Dirichlet condition $u_{\boldsymbol{n}} = 0$  weakly on $\Gamma_{bg}$. It can be considered as a penalty term. Since $\boldsymbol{u} = u_{\boldsymbol{n}}\boldsymbol{n} + u_{\boldsymbol{t}}\boldsymbol{t}$, the contribution of the tangential force can also be written $\beta\boldsymbol{u}\cdot\boldsymbol{v}$ when $u_{\boldsymbol{n}} = 0$. The value of the positive parameter $\gamma_0$ depends on the physical problem and $h$ is a measure of the mesh size on $\Gamma_b$. The sensitivity of the GL positions for different values of $\gamma_0$ is shown in Sect. 5. The first term in $B_\mathcal{N}$ symmetrizes the boundary term $B_\Gamma + B_\mathcal{N}$ on $\Gamma_{bg}$ and vanishes when $u_{\boldsymbol{n}} = 0$. The boundary term $F_\Gamma(\boldsymbol{v})$ is from the buoyancy force at the ice/ocean interface in (6) where $p_w$ depends on $z_b$ on $\Gamma_{bf}$.

## 3.2 The discretized FS equations

We employ linear Lagrange elements with Galerkin Least Square (GLS) stabilization (Franca and Frey, 1992; Helanow and Ahlkrona, 2018) to avoid spurious oscillations in the pressure using the standard MISMIP setting in Elmer/ICE (Durand et al., 2009a; Gagliardini et al., 2013) approximating solutions in the spaces $V_k$ and $Q_{k*}$ in Eq. (13).

The mesh is constructed from a footprint mesh on the ice base and then extruded with the same number of layers equidistantly in the vertical direction according to the thickness of the ice sheet. To simplify the implementation in 2D, the footprint mesh on the ice base consists of $N+1$ nodes at $\boldsymbol{x}_i = (x_i, z_b(x_i))$, $i = 0, \ldots, N$, with $x$-coordinates $x_i$ and a constant mesh size $\Delta x = x_i - x_{i-1}$.

In general, the GL is somewhere in the interior of an interval $[x_{i-1}, x_i]$ and it crosses the interval boundaries as it moves forward in the advance phase and backward in the retreat phase of the ice. The advantage with Nitsche's method of formulating the boundary conditions is that if $x_{GL} \in [x_{i-1}, x_i]$ then the boundary integral over the interval can be split into two parts in Eq. (14) such that $(x, z_b(x)) \in \Gamma_{bg}$ when $x \in [x_{i-1}, x_{GL}]$ and if $x \in [x_{GL}, x_i]$ then $(x, z_b(x)) \in \Gamma_{bf}$. In the GL element, we have

$$
\begin{aligned}
B_\Gamma + B_\mathcal{N} &= \int\limits_{[x_{i-1}, x_{GL}]} -(\sigma_{\boldsymbol{nn}}(\boldsymbol{u}, p)\boldsymbol{n} \cdot \boldsymbol{v} + \sigma_{\boldsymbol{nn}}(\boldsymbol{v}, q)\boldsymbol{n} \cdot \boldsymbol{u}) + \beta(\boldsymbol{t} \cdot \boldsymbol{u})(\boldsymbol{t} \cdot \boldsymbol{v}) + \frac{\gamma_0}{h}(\boldsymbol{n} \cdot \boldsymbol{u})(\boldsymbol{n} \cdot \boldsymbol{v}) \ \mathrm{d}s, \\
F_\Gamma &= -\int\limits_{[x_{GL}, x_i]} p_w \boldsymbol{n} \cdot \boldsymbol{v} \ \mathrm{d}s,
\end{aligned}
\tag{15}
$$

with the integration element $\mathrm{d}s$ following $\Gamma_b$. There is a change of the boundary condition in the middle of the FEM element where the GL is located. With a strong formulation of the boundary condition $u_{\boldsymbol{n}} = 0$, the basis functions in $V_k$ share this property and the condition changes from the grounded node $x_{i-1}$ where the basis function satisfies $u_{\boldsymbol{n}} = 0$ to the floating node at $x_i$ with a free $u_{\boldsymbol{n}}$ without taking the position of the GL inside $[x_{i-1}, x_i]$ into account. With the weak formulation in Nitsche's method, the standard basis functions we use do not satisfy $u_{\boldsymbol{n}} = 0$ strictly. The boundary condition is imposed on the solution by the additional penalty term multiplied by $\gamma_0$ in $B_\mathcal{N}$ in (14). A large $\gamma_0$ will force $u_{\boldsymbol{n}}$ to be small. The penalty term may change inside an element as in (15) where it is $\neq 0$ only in the grounded part.

The resulting system of nonlinear equations form a nonlinear complementarity problem (Christensen et al., 1998). The distance $d$ between the base of the ice and the bedrock at time $t$ and at $x$ is

$$
d(x, t) = z_b(x, t) - b(x) \geq 0.
\tag{16}
$$

If $d > 0$ on $\Gamma_{bf}$ then the ice is not in contact with the bedrock and $\sigma_{\boldsymbol{nn}} + p_w = 0$ and if $\sigma_{\boldsymbol{nn}} + p_w < 0$ on $\Gamma_{bg}$ then the ice and the bedrock are in contact and $d = 0$. Hence, the complementarity relation in the vertical direction is

$$
d(x, t) \geq 0, \ \sigma_{\boldsymbol{nn}} + p_w \leq 0, \quad d(x, t)(\sigma_{\boldsymbol{nn}} + p_w) = 0 \text{ on } \Gamma_b.
\tag{17}
$$

The contact friction law is such that $\beta > 0$ when $x < x_{GL}$ and $\beta = 0$ when $x > x_{GL}$. The complementarity relation along the ice base at $x$ is then the non-negativity of $d$ and

$$
\beta \geq 0, \ \beta(x, t)d(x, t) = 0 \text{ on } \Gamma_b.
\tag{18}
$$

In particular, these relations are valid at the nodes $x = x_j$, $j = 0, 1, \ldots, N$.

The complementarity condition also holds for $u_{\boldsymbol{n}}$ and $\sigma_{\boldsymbol{nn}}$ such that

$$\sigma_{\boldsymbol{nn}} + p_w \leq 0, \quad u_{\boldsymbol{n}}(\sigma_{\boldsymbol{nn}} + p_w) = 0 \text{ on } \Gamma_b, \tag{19}$$

without any sign constraint on $u_{\boldsymbol{n}}$ except for the retreat phase when the ice leaves the ground and $u_{\boldsymbol{n}} < 0$.

     Similar implementations for contact problems using Nitsche's method are found in Chouly et al. (2017a, b), where the unknowns in the PDEs are the displacement fields instead of the velocity in Eq. (1). Analysis in Chouly et al. (2017a) suggests

that Nitsche's method for the contact problem can provide a stable numerical solution with an optimal convergence rate.

     The nonlinear equations, Eq. (14), for the nodal values of $\boldsymbol{u}$ and $p$ are solved by Picard iterations. The system of linear equations in every Picard iteration is solved directly by using the MUMPS linear solver in Elmer/ICE. The condition on $d_j = d(x_j)$ is used to decide if the node $x_j$ is geometrically grounded or floating. It is computed at each time step and is not changed during the nonlinear iterations (Picard). The procedure for solution of the nonlinear FS equations is outlined in

Algorithm 1. In two dimensions, the GL will be located in one element.

---

**Algorithm 1** Solve the FS equations

---

For a given mesh, compute $d_j$, $j = 0, 1, \ldots, N$, for all the nodes $x_j$ at the ice base.

Mark node $j$ as geometrically grounded if $d_j < 10^{-3}$, otherwise floating.

Find the element which contains both geometrically grounded and floating nodes, and mark the grounded node in this element as 'GL node'.

Compute the residual of the FS equations with the initial guess of the solution.

**while** the residual is larger than the tolerance **do**

     Assemble the FEM matrix for the interior of the domain $\Omega$.

     **for** the boundary elements on $\Gamma_b$ **do**

         **if** has 'GL node' **then**

             Mark the current element as a 'potential GL element'.

             Use the subgrid scheme in Algorithm 3 of Sect. 4 for the assembly.

         **else**

             Assemble the boundary element.

         **end if**

     **end for**

     Solve the linearized FS equations for a correction of the solution.

     Compute the solution and the residual.

**end while**

---

## 3.3 Discretization of the advection equations

The advection equations for the moving ice boundary in Eq. (7) and (8) are discretized in time by a finite difference method and in space by FEM with linear Lagrange elements for $z_s$ and $z_b$. An artificial diffusion stabilization term is added, making the spatial discretization behave like an upwind scheme in the direction of the velocity as implemented in Elmer/ICE.

The advection equations Eq. (7) and Eq. (8) are integrated in time by a semi-implicit method of first order accuracy. Let $c = s$ or $b$. Then the solution is advanced from time $t^\ell$ to $t^{\ell+1} = t^\ell + \Delta t$ with the time step $\Delta t$ by

$$z_c^{\ell+1} = z_c^\ell + \Delta t (a_c^\ell - u_c^\ell \frac{\partial z_c^{\ell+1}}{\partial x} + w_c^\ell). \tag{20}$$

The spatial derivative of $z_c$ is approximated by FEM as described above. A system of linear equations is solved at $t^{\ell+1}$ for $z_c^{\ell+1}$. This time discretization and its properties are discussed in Cheng et al. (2017) and summarized in Algorithm 2.

---

**Algorithm 2** Time scheme of the GL migration problem

---

Start from an initial geometry $\Omega^0$ defined by $z_b^0, z_s^0$.

**for** $\ell = 0$ to $T/\Delta t - 1$ **do**

    Solve the FS equations on $\Omega^\ell$ with Algorithm 1, to get the solution $\boldsymbol{u}^\ell$.

    Solve for $z_b^{\ell+1}$ and $z_s^{\ell+1}$ with $\boldsymbol{u}^\ell$ by the semi-implicit Euler method.

    Use $z_b^{\ell+1}$ and $z_s^{\ell+1}$ to update $\Omega^{\ell+1}$.

**end for**

---

A numerical stability problem in $z_b$ is encountered in the boundary condition at $\Gamma_{bf}$ when the FS equations are solved in Durand et al. (2009a). It is resolved by expressing $z_b$ in $p_w$ at $\Gamma_{bf}$ with a damping term. An alternative interpretation of the idea in Durand et al. (2009a) and an explanation follow below.

The relation between $u_{\boldsymbol{n}}$ and $u_{\boldsymbol{t}}$ at $\Gamma_{bf}$ and $\boldsymbol{u}_b = \boldsymbol{u}(x, z_b(x))$ is

$$\boldsymbol{u}_b = \begin{pmatrix} u_b \\ w_b \end{pmatrix} = \begin{pmatrix} z_{bx} \\ -1 \end{pmatrix} \frac{u_{\boldsymbol{n}}}{\sqrt{1 + z_{bx}^2}} + \begin{pmatrix} 1 \\ z_{bx} \end{pmatrix} \frac{u_{\boldsymbol{t}}}{\sqrt{1 + z_{bx}^2}}, \tag{21}$$

where $z_{bx}$ denotes $\partial z_b / \partial x$. Inserting $u_b$ and $w_b$ from Eq. (21) into Eq. (8) yields

$$\frac{\partial z_b}{\partial t} = a_b - u_{\boldsymbol{n}} \sqrt{1 + z_{bx}^2}. \tag{22}$$

Instead of discretizing Eq. (22) explicitly at $t^{\ell+1}$ with $u_{\boldsymbol{n}}^\ell$ to determine $p_w^{\ell+1}$, the base coordinate is updated implicitly

$$z_b^{\ell+1} = z_b^\ell + \Delta t \left( a_b^{\ell+1} - u_{\boldsymbol{n}}^{\ell+1} \sqrt{1 + (z_{bx}^{\ell+1})^2} \right) \tag{23}$$

in the evaluation of $p_w$ in $F_\Gamma(\boldsymbol{v})$ in Eq. (14).

Assuming that $z_{bx}$ is small, the time step restriction in Eq. (23) is estimated by considering a 2D slab of the floating ice of width $\Delta x$ and thickness $H$. Newton's law of motion yields

$$M \dot{u}_{\boldsymbol{n}} = Mg - \Delta x p_w,$$

where $M = \Delta x(z_s - z_b)\rho$ is the mass of the slab. Dividing by $M$, integrating in time for $u_{\boldsymbol{n}}(t^m)$, letting $m = \ell + 1$ or $\ell$, and approximating the integral by the trapezoidal rule for the quadrature yields

$$240 \quad u_{\boldsymbol{n}}(t^m) = \int\limits_0^{t^m} g + \frac{g\rho_w}{\rho}\frac{z_b}{z_s - z_b}\,\mathrm{d}s \approx gt^m + \frac{g\rho_w}{\rho}\sum_{i=0}^{m}\alpha_i \frac{z_b^i}{z_s^i - z_b^i}\Delta t = u_{\boldsymbol{n}}^m,$$

with the parameters

$$\alpha_i = 0.5,\ i = 0, m, \quad \alpha_i = 1,\ i = 1,\ldots,m-1.$$

Then insert $u_{\boldsymbol{n}}^m$ into Eq. (23). All terms in $u_{\boldsymbol{n}}^m$ from time steps $i < m$ are collected in the sum $\Delta t F^{m-1}$. Then Eq. (23) can be written

$$245 \quad z_b^{\ell+1} = z_b^\ell - \Delta t^2 \frac{g\rho_w}{2\rho}\frac{z_b^m}{z_s^m - z_b^m} + \Delta t\left(a_b^\ell - gt^m - \Delta t F^{m-1}\right). \tag{24}$$

For small changes in $z_b$ in Eq. (24), the explicit method with $m = \ell$ is stable when $\Delta t$ is so small that

$$|1 - \Delta t^2 \frac{g\rho_w}{2H\rho}| \leq 1. \tag{25}$$

When $H = 100$ m on the ice shelf, $\Delta t < 6.1$ s which is far smaller than the stable steps for Eq. (20). Choosing the implicit scheme with $m = \ell + 1$, the bound on $\Delta t$ is

$$250 \quad 1/|1 + \Delta t^2 \frac{g\rho_w}{2H\rho}| \leq 1, \tag{26}$$

i.e. there is no bound on positive $\Delta t$ for stability but accuracy will restrict $\Delta t$.

Much longer stable time steps are possible at the surface and the base of the ice with a semi-implicit method Eq. (20) and a fully implicit method Eq. (23) compared to an explicit method. For example, the time step for the problem in Eq. (20) with 1 km mesh size can be up to a couple of months. Therefore, we use the scheme in Eq. (20) for Eqs. (7) and (8) and the scheme in Eq. (23) for Eq. (22) and $p_w$ as in Durand et al. (2009a). The difference between the approximations of $z_b$ in Eq. (20) and (23) is of $\mathcal{O}(\Delta t^2)$.

## 4 Subgrid scheme around the grounding line

The basic idea of the subgrid scheme for the FS equations in this paper follows the GL parameterization (SEP3) for SSA in Seroussi et al. (2014) and the analysis for FS in Schoof (2011). The GL is located at the position where the ice is on the ground and the flotation criterion is perfectly satisfied such that $\sigma_{\boldsymbol{nn}} = -p_w$. In the FS equations, the hydrostatic assumption Eq. (9) may not be valid close to the GL. Therefore, the GL position can not be determined by simply checking the total thickness of the ice $H$ against the depth below sea level $H_{bw}$. Instead, the flotation criterion is computed by comparing the water pressure with the numerical normal stress component orthogonal to the boundary inspired by the first order analysis in Sect. 2.4.

The numerical solutions, e.g. Gagliardini et al. (2016); Gladstone et al. (2017), converge to the analytical solution of the FS PDE as the mesh size decreases. The analytical solution satisfies $z_b(x,t) > b(x)$ with the boundary conditions in Eq. (6) at the

base of the floating ice, and where the ice is in contact with the bedrock $z_b(x,t) = b(x)$, the boundary conditions are given by Eq. (5). Examples of the analytical solution are demonstrated by the thin light blue lines in Figs. 2 and 3 with a black '∗' at the analytical GL position $x_{GL}$. The two figures share the same analytical solution. However, as illustrated in Figs. 2 and 3, the basal boundary of the ice $z_b(x,t)$ does not conform with the mesh from the spatial discretization. In particular, the GL position $x_{GL}$ of the analytical solution does not coincide with any of the nodes, but it usually stays on the bedrock $b(x)$ between the last grounded ($x_{i-1}$) and the first floating ($x_i$) nodes, see Figs. 2 and 3. The linear element boundary between any $x_{j-1}$ and $x_j$ is denoted by $\mathcal{E}_j$. The sequence of $\mathcal{E}_j, j = 1, \ldots, N$, approximates $\Gamma_b$. The grounding line element containing the GL is $\mathcal{E}_i$.

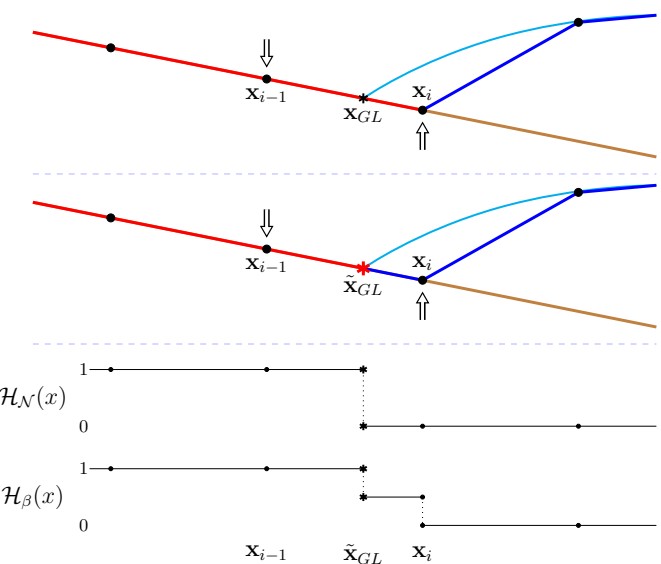

**Figure 2.** Schematic figure of the GL in case i, with the arrows indicating the direction of the net forces in the vertical direction. The light blue line is the analytical solution of the ice sheet with the analytical GL position $x_{GL}$. The red line is the grounded boundary $\Gamma_{bg}$, the dark blue line is the floating boundary $\Gamma_{bf}$, and the brown line is the bedrock topography $b(x)$. Upper panel: The last grounded and first floating nodes as defined in Elmer/ICE. Middle panel: Linear interpolation to approximate the numerical GL position $\tilde{x}_{GL}$. Lower panel: The step functions $\mathcal{H}_\mathcal{N}(x)$ and $\mathcal{H}_\beta(x)$ indicate the area for Nitsche's penalty and slip boundary conditions.

Depending on how the mesh is created from the initial geometry and updated during the simulation, the first floating node at $x_i$, as well as the GL element, can be either on the bedrock (as in Fig. 2) or at the ice base above the bedrock (as in Fig. 3), even though the corresponding analytical solutions are identical. Denote the situation in Fig. 2 as case i, and the one in Fig. 3 as case ii. The physical boundary conditions of the two cases are different only at the GL element. More precisely, in case i, the net force in the vertical direction on the node $x_i$ is pointing inward, namely $\chi(x_i) = \sigma_{nn}(x_i) + p_w(x_i) > 0$, whereas in case ii, the floating condition $\sigma_{nn}(x_i) + p_w(x_i) = 0$ is satisfied in the node $x_i$. The directions of the vertical net force at the nodes $x_{i-1}$ and $x_i$ are shown by the arrows in the upper panels of Fig. 2 and 3. Consequently, the external forces and boundary conditions imposed on the GL element are different in the two cases. For instance, in case i, the GL element is considered

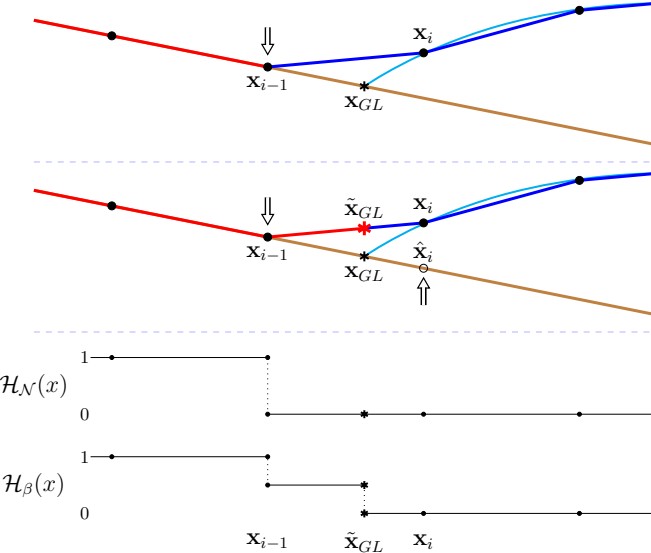

**Figure 3.** Schematic figure of the GL in case ii, with the arrows indicating the direction of the net forces in the vertical direction. The colors of the lines follow those in Fig. 2. Upper panel: The last grounded and first floating nodes as defined in Elmer/ICE. The node $\boldsymbol{x}_i$ is fully geometrically floating and the net force is 0. Middle panel: Linear interpolation to approximate the numerical GL position $\tilde{\boldsymbol{x}}_{GL}$. The point $\hat{\boldsymbol{x}}_i$ on the bedrock has the same $x$ coordinate as $\boldsymbol{x}_i$. Lower panel: The step functions $\mathcal{H}_{\mathcal{N}}(x)$ and $\mathcal{H}_{\beta}(x)$ indicate the area for Nitsche's penalty and slip boundary conditions.

as geometrically grounded (defined as in Algorithm 1), shown with red color in the upper panel of Fig. 2. In case ii, the GL element is treated as geometrically floating and colored in blue in the upper panel of Fig. 3.

These two cases are similar to the LG and FF cases in Gagliardini et al. (2016) implying that the numerical solutions in the two cases are different, especially on a coarse mesh (mesh size at about 100 m or larger). Thus, we propose a subgrid scheme

to reduce these differences in the spatial discretization and to capture the GL migration without using a fine mesh resolution ($< 100$ m). The schematic drawing of the subgrid scheme for the two cases is shown in the middle panels of Fig. 2 and 3. The GL element is divided into the grounded (red) and floating (blue) parts by the estimated GL position $\tilde{\boldsymbol{x}}_{GL}$ on $\mathcal{E}_i$, which is the numerical approximation of the analytical GL position $\boldsymbol{x}_{GL}$.

The GL moves toward the ocean in the advance phase and away from the ocean in the retreat phase. First, we consider case

i in the *advance phase* and define the indicator by

$$\chi(\boldsymbol{x}) = \sigma_{\boldsymbol{nn}} + p_w, \tag{27}$$

which vanishes on the floating ice and is negative and approximately equal to $\chi_a = \tau_{22} - p + p_w$ in Eq. (11) on the ground since the slope of the bedrock is small and $\boldsymbol{n} \approx (0, -1)^T$. Because of the poor spatial resolution of the coarse mesh, $\chi(\boldsymbol{x}_i)$ is positive.

To determine the position $\tilde{\boldsymbol{x}}_{GL}$, we solve $\chi(\tilde{\boldsymbol{x}}_{GL}) = \sigma_{\boldsymbol{nn}}(\tilde{\boldsymbol{x}}_{GL}) + p_w(\tilde{\boldsymbol{x}}_{GL}) = 0$ by linear interpolation between $\chi(\boldsymbol{x}_{i-1})$ and $\chi(\boldsymbol{x}_i)$ such that

$$\tilde{\boldsymbol{x}}_{GL} = \boldsymbol{x}_{i-1} - \frac{\chi(\boldsymbol{x}_{i-1})}{\chi(\boldsymbol{x}_{i-1}) - \chi(\boldsymbol{x}_i)}(\boldsymbol{x}_{i-1} - \boldsymbol{x}_i). \tag{28}$$

The water pressure $p_w(\boldsymbol{x})$ is a linear function of $\boldsymbol{x}$ on the GL element and the numerical solution of $\sigma_{\boldsymbol{nn}}(\boldsymbol{x})$ is also piecewise linear on every element with the standard Lagrange elements in Elmer/ICE (Gagliardini et al., 2013). Hence, it makes sense to approximate the analytical GL position $\boldsymbol{x}_{GL}$ by $\tilde{\boldsymbol{x}}_{GL}$ by linear interpolation in the current framework. This approach fits well with case i since the indicator $\chi(\boldsymbol{x})$ has opposite signs at $\boldsymbol{x}_{i-1}$ and $\boldsymbol{x}_i$, see the middle panel of Fig. 2 where $\tilde{\boldsymbol{x}}_{GL}$ is marked by a red '∗'. It guarantees the existence and uniqueness of $\tilde{\boldsymbol{x}}_{GL}$ on the GL element.

Another situation in the advance phase is case ii shown in Fig. 3. As the elements on both sides of the node $\boldsymbol{x}_i$ are geometrically floating, the boundary condition imposed on $\boldsymbol{x}_i$ becomes $\chi(\boldsymbol{x}_i) = \sigma_{\boldsymbol{nn}}(\boldsymbol{x}_i) + p_w(\boldsymbol{x}_i) = 0$. However, the implicit treatment of the ice base moves the $z$-coordinate of the node $\boldsymbol{x}_i$ towards the bedrock with $u_{\boldsymbol{n}} > 0$ in Eq. (23) as discussed in Sect. 3.3. The result is that $p_w$ defined by the implicit $z_b$ in (23) satisfies $\sigma_{\boldsymbol{nn}} + p_w > 0$ in (27) and $\chi(\boldsymbol{x}_i) > 0$.

The implicit treatment of the ice base has the consequence that only case ii occurs in the *retreat phase*. When the FS equations are solved, the implicit update of the ice base with $u_{\boldsymbol{n}} < 0$ in Eq. (23) implies that the last grounded node in the previous time step is leaving the bedrock when the ice is retreating and the GL moves back to the adjacent element. Case i will not appear in that situation since $z_b(x_i) > b(x_i)$. In this circumstance, $\chi(\boldsymbol{x}_i) = 0$ in the floating node and a correction of $\chi(\boldsymbol{x})$ is introduced into case ii by $\tilde{\chi}$ in

$$\tilde{\chi}(\boldsymbol{x}) = \sigma_{\boldsymbol{nn}}(\boldsymbol{x}) + p_b(\boldsymbol{x}). \tag{29}$$

Here $p_b(\boldsymbol{x}) = -\rho_w g b(x)$ is the water pressure on the bedrock corresponding to linear extrapolation of the pressure for $x > x_{GL}$ along the element on the bedrock. Furthermore, $\tilde{\chi}(\boldsymbol{x}) \geq \chi(\boldsymbol{x})$. Notice that $p_b(\boldsymbol{x}_i) = p_w(\hat{\boldsymbol{x}}_i) > p_w(\boldsymbol{x}_i)$, where $\hat{\boldsymbol{x}}_i$ is a point on the bedrock with the same $x$ coordinate of $\boldsymbol{x}_i$, as illustrated in the middle panel of Fig. 3. Both $\chi(\boldsymbol{x})$ in (27) and $\tilde{\chi}(\boldsymbol{x})$ in (29) are nonlinear in $\boldsymbol{x}$ but the numerical approximation of them will vary linearly in $\boldsymbol{x}$. A solution $\tilde{\boldsymbol{x}}_{GL}$ is found by linear interpolation of $\tilde{\chi}(\boldsymbol{x})$ between the nodes $\boldsymbol{x}_{i-1}$ and $\boldsymbol{x}_i$ as in Eq. (28). It follows from Eq. (28) that $\tilde{\boldsymbol{x}}_{GL}$ is located on the element boundary, see Figs. 2 and 3. If we compare with case i, this correction can be considered as using $\sigma_{\boldsymbol{nn}}(\tilde{\boldsymbol{x}}_{GL})$ to approximate $\sigma_{\boldsymbol{nn}}(\boldsymbol{x}_{GL})$ on a virtual element between $\boldsymbol{x}_{i-1}$ and $\hat{\boldsymbol{x}}_i$, see Fig. 3. The position $\tilde{\boldsymbol{x}}_{GL}$ is a numerical approximation of the analytical GL position, although it is not geometrically in contact with the bedrock.

Since we have $p_b(\boldsymbol{x}) = p_w(\boldsymbol{x})$ and $\chi(\boldsymbol{x}) = \tilde{\chi}(\boldsymbol{x})$ at the GL element in case i, we can simply use $\tilde{\chi}(\boldsymbol{x})$ to find $\tilde{\boldsymbol{x}}_{GL}$ for the two cases by replacing $\chi$ in (28) by $\tilde{\chi}$.

The domains $\Gamma_{bg}$ and $\Gamma_{bf}$ are separated at $\tilde{\boldsymbol{x}}_{GL}$ as in Eq. (15) and the integrals on the GL element are calculated with a high-order integration scheme as in Seroussi et al. (2014). We introduce two step functions $\mathcal{H}_{\mathcal{N}}(x)$ and $\mathcal{H}_{\beta}(x)$ to include and exclude quadrature points in the integration of Nitsche's term and the slip boundary condition, respectively. They are defined for case i in Fig. 2 and for case ii in Fig. 3. To achieve a reasonable numerical accuracy within the GL element, as suggested in Seroussi et al. (2014), at least tenth order Gaussian quadrature is used.

The penalty term in Nitsche's method restricts the motion of the element in the normal direction. It is only imposed on an element which is fully geometrically on the ground in case i. On the contrary in case ii, the GL element $\mathcal{E}_i$ is not in contact with the bedrock, see Fig. 3. The normal velocity on the element should not be forced to zero and only the floating boundary condition is then used on the GL element. Nitsche's penalty term should be imposed on all the fully geometrically grounded elements and partially on the GL element in the advance phase as in case i. The step function $\mathcal{H}_\mathcal{N}(x)$ indicates how Nitsche's method is implemented on the basal elements, see the lower panels of Fig. 2 and 3 for the two cases. The penalty term contributes to the integration only when $\mathcal{H}_\mathcal{N}(x) = 1$.

The slip coefficient $\beta$ is treated similarly with the step function $\mathcal{H}_\beta(x)$, where $\mathcal{H}_\beta(x) = 1$ is on the fully geometrically grounded elements and $\mathcal{H}_\beta(x) = 0$ on the floating elements. To further smooth the transition of $\beta$ at the GL, the step function is set to be 1/2 in parts of the GL element before integrating using the high order scheme. The convergence of the nonlinear iterations is improved in this way. In case i, full friction is applied at the grounded part between $\boldsymbol{x}_{i-1}$ and $\tilde{\boldsymbol{x}}_{GL}$ of the GL element since this part is also geometrically grounded in the analytical solution of the FS as in Fig. 2. Then, the friction is lower in the remaining part of $\mathcal{E}_i$. For the floating part between $\tilde{\boldsymbol{x}}_{GL}$ and $\boldsymbol{x}_i$ in case ii, there is no friction and $\mathcal{H}_\beta(x) = 0$ and we have reduced friction between $\boldsymbol{x}_{i-1}$ and $\tilde{\boldsymbol{x}}_{GL}$, see the lower panel of Fig. 3. The boundary integral Eq. (15) on $\mathcal{E}_i$ is now rewritten with the two step functions as

$$B_\Gamma + B_\mathcal{N} = \int_{\mathcal{E}_i} -\mathcal{H}_\mathcal{N}(\sigma_{\boldsymbol{nn}}(\boldsymbol{u},p)\boldsymbol{n}\cdot\boldsymbol{v} + \sigma_{\boldsymbol{nn}}(\boldsymbol{v},q)\boldsymbol{n}\cdot\boldsymbol{u}) + \mathcal{H}_\beta\beta(\boldsymbol{t}\cdot\boldsymbol{u})(\boldsymbol{t}\cdot\boldsymbol{v}) + \mathcal{H}_\mathcal{N}\frac{\gamma_0}{h}(\boldsymbol{n}\cdot\boldsymbol{u})(\boldsymbol{n}\cdot\boldsymbol{v}) \ \mathrm{d}s,$$

$$F_\Gamma = \int_{\mathcal{E}_i} (1 - \mathcal{H}_\mathcal{N})p_w\boldsymbol{n}\cdot\boldsymbol{v} \ \mathrm{d}s. \tag{30}$$

A summary of the numerical treatment of the GL is:

- Advance phase $\Rightarrow$ indicator $\chi$ in (27), case i or case ii

- Retreat phase $\Rightarrow$ indicator $\tilde{\chi}$ in (29), case ii

The case is determined by the geometry of the GL element and the sign of the indicator $\chi$.

The algorithm for the GL element is:

---
**Algorithm 3** Subgrid modeling for the GL element
---

Take all the 'potential GL elements' and solve $\chi(\boldsymbol{x}) = 0$ (advance phase) or $\tilde{\chi}(\boldsymbol{x}) = 0$ (retreat phase) to find $\tilde{\boldsymbol{x}}_{GL}$ and the GL element.

Determine which case this GL element belongs to by checking the geometrical conditions at $\boldsymbol{x}_i$.

Specify $\mathcal{H}_\mathcal{N}(x)$ and $\mathcal{H}_\beta(x)$ based on $\tilde{\boldsymbol{x}}_{GL}$ depending on the case and the advance or retreat phase.

Integrate Eq. (30) for the FEM matrix assembly.

---

Equations (1), (7), and (8) form a system of coupled nonlinear equations. They are solved by Elmer/ICE v.8.3 in the same manner as Durand et al. (2009b); Gagliardini et al. (2013, 2016). The detailed procedure is explained in Algorithms 1, 2, and 3. The solution to the nonlinear FS system is computed with Picard iterations to a $10^{-5}$ relative error with a limit of maximal

25 nonlinear iterations. The $\tilde{x}_{GL}$ position is determined dynamically during each fixed-point iteration by solving Eq. (28) with $\chi$ or $\tilde{\chi}$ and the solution $\sigma_{nn}(x)$ from the previous nonlinear iteration, and the step functions $\mathcal{H}_{\mathcal{N}}$ and $\mathcal{H}_{\beta}$ are adjusted accordingly. The water pressure $p_b$ is fixed since the ice geometry is not changed during the nonlinear iterations.

## 5 Results

The numerical experiments follow the MISMIP benchmark (Pattyn et al., 2012) and a comparison is made with the results in Gagliardini et al. (2016). Using the experiment MISMIP 3a, the setups are exactly the same as in the advancing and retreating simulations in Gagliardini et al. (2016). The experiments are run with spatial resolutions of $\Delta x = 4$ km, 2 km, 1 km and 0.5 km. The mesh at the base is extruded vertically in 20 layers with equidistantly placed nodes in each vertical column. The time step is $\Delta t = 0.125$ year for all four resolutions to eliminate time discretization errors when comparing different spatial resolutions.

The dependence on $\gamma_0$ in (30) for the retreating ice is shown in Fig. 4 with $\gamma_0$ between $10^4$ and $10^9$. The estimated GL positions do not vary with different choices of $\gamma_0$ from $10^5$ to $10^8$ which suggests a suitable range of $\gamma_0$. If $\gamma_0$ is too small ($\gamma_0 \ll 10^4$), oscillations appear in the estimated GL positions. If $\gamma_0$ is too large ($\gamma_0 \gg 10^8$), then more nonlinear iterations in Algorithm 1 are needed in each time step. The same dependency of $\gamma_0$ is observed for the advancing experiments and for different mesh resolutions as well. The results are not very sensitive to $\gamma_0$ and for the remaining experiments we choose $\gamma_0 = 10^6$.

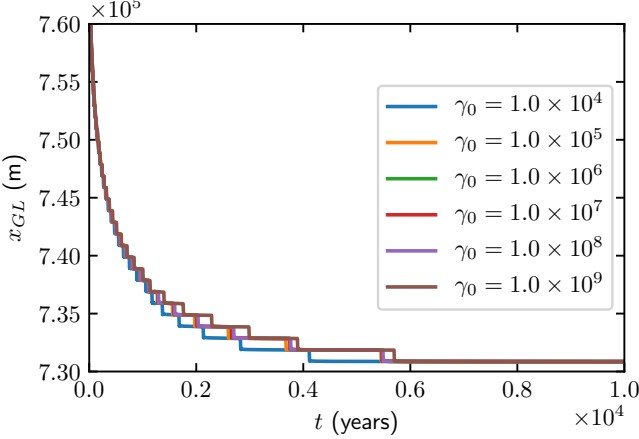

**Figure 4.** The MISMIP 3a retreat experiment with $\Delta x = 1$ km for different choices of $\gamma_0$ in the time interval $[0, 10000]$ years.

The GL position during the transient simulations in the advance and retreat phases are displayed in Fig. 5 and the steady state results (at $t = 10000$) are shown in Fig. 6 for different mesh resolutions. The range of the steady state solutions from Gagliardini et al. (2016) with mesh resolution from 25 m to 200 m are shown as background shaded regions in red. We achieve similar GL migration results for both the advance and retreat experiments with at least 20 times larger mesh resolutions. The GL position is insensitive to the variation in mesh size between 0.5 km and 4 km.

The distance between the steady state GL positions of the retreat and the advance phases is shown in Fig. 6 (b). The maximal distance is about 6 km at $\Delta x = 1$ km with the subgrid model, whereas in Gagliardini et al. (2016), the resolution has to be below 50 m to achieve a similar result.

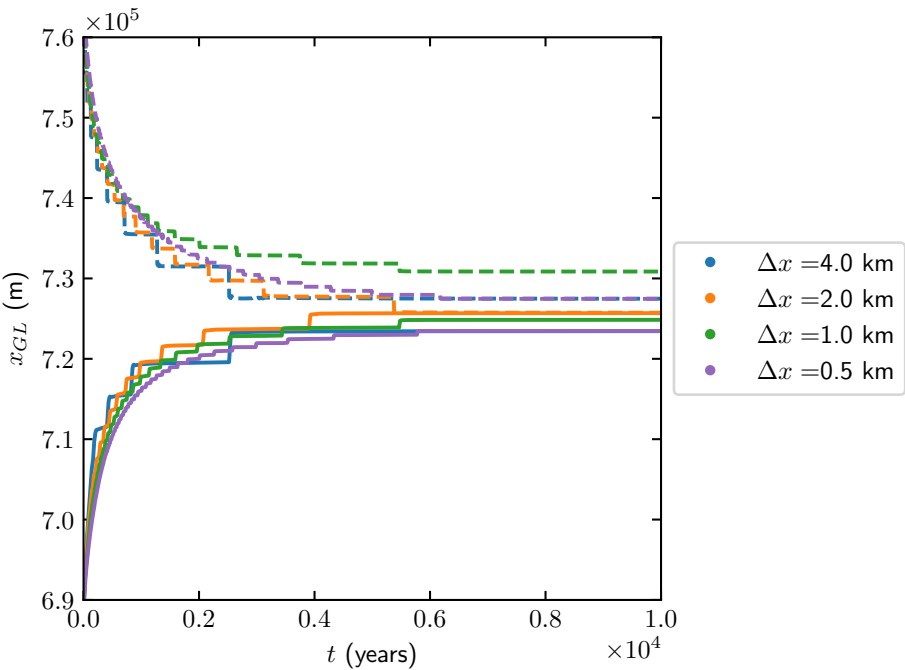

**Figure 5.** The MISMIP 3a experiments for the GL position when $t \in [0, 10000]$ with $\Delta x = 4$ km, 2 km, 1 km and 0.5 km for the advance (solid) and retreat (dashed) phases.

We observe oscillations at the ice surface near the GL in all the experiments as expected from Durand et al. (2009a); Schoof (2011). A zoom-in plot of the surface elevation computed with four different mesh sizes $\Delta x = 0.5, 1, 2, 4$ km after an advance simulation to $t = 10000$ years is found to the left in Fig. 7. The abscissa is the distance from the steady state GL position for each mesh size. In general, the estimated GL position does not coincide with any nodes even at the steady state but it may be close to a node.

The ratio between the thickness below sea level $H_{bw}$ and the ice thickness $H$ is shown to the right in Fig. 7. As in the left panel, the ratio is plotted versus the distance from the GL achieved with the particular mesh size. The horizontal, purple, dash-dotted line represents the ratio of $\rho/\rho_w$. The solutions vary smoothly over the mesh with $\Delta x = 0.5$ km which appears to be a sufficient resolution in both panels of Fig. 7. Moreover, the solutions converge regularly toward the solution with $\Delta x = 0.5$ km when the mesh size decreases.

The result for $\Delta x = 0.5$ km in the right panel confirms that the hydrostatic assumption $H\rho = H_{bw}\rho_w$ in Eq. (9) is not valid in the FS equations for $x > x_{GL}$ close to the GL and at the GL position, cf. Durand et al. (2009a); Schoof (2011). For $x < x_{GL}$ we

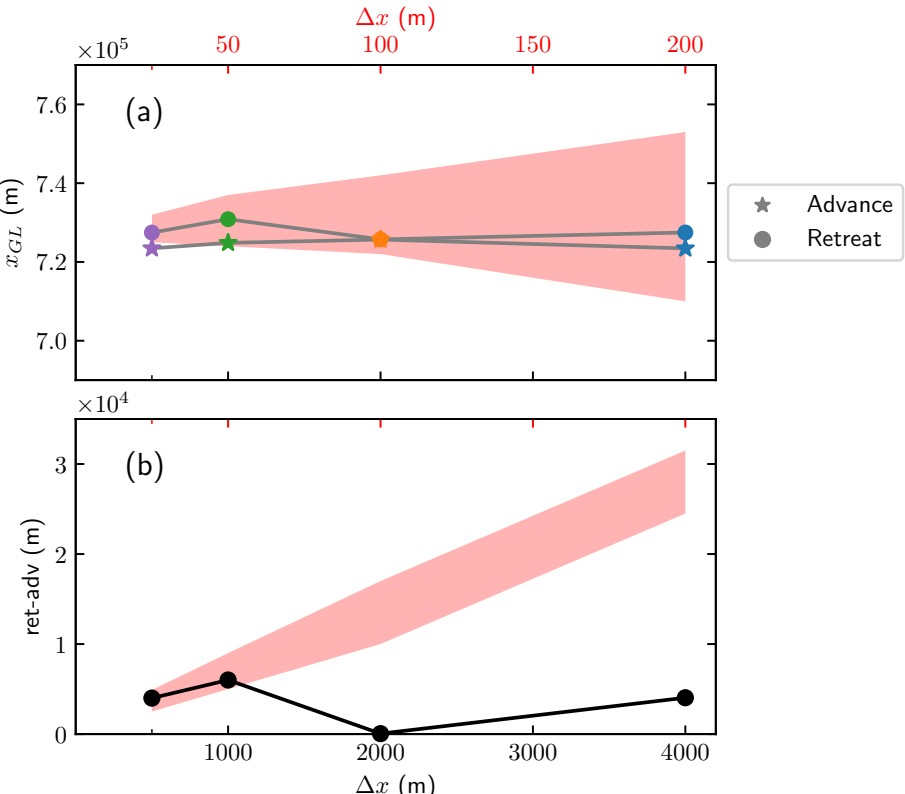

**Figure 6.** The MISMIP 3a experiments at the final time $t = 10000$ with the resolutions at $\Delta x = 4$ km, 2 km, 1 km and 0.5 km. (a) The GL positions in the advance (★) and retreat (●) phases. (b) The distance between the retreat and the advance $x_{GL}$ at the steady states. The shaded regions indicate the range of the results in Gagliardini et al. (2016) with *20 times smaller mesh resolutions* from 25 to 200 m with the axis scale shown in red at the top of the plot.

have that $H_{bw}/H < \rho/\rho_w$ since $H_{bw}$ decreases and $H$ increases. The conclusion from numerical experiments in van Dongen et al. (2018) is that the hydrostatic assumption and the SSA equations approximate the FS equations well for the floating ice beginning at a short distance away from the GL.

The surface and the base velocity solutions from the advance experiment are displayed in Fig. 8 with $\Delta x = 0.5, 1, 2, 4$ km after 10000 years. The horizontal velocities on the two surfaces are on top of each other for all $\Delta x$ with negligibly small differences on the floating ice as expected. The vertical velocities $w$ on the surface (dotted curves) and the base (solid curves) at the GL are almost discontinuous as analyzed in Schoof (2011). With the subgrid model, the rapid variation is captured with $\Delta x = 0.5$ km. The convergence for decreasing mesh size behaves smoothly.

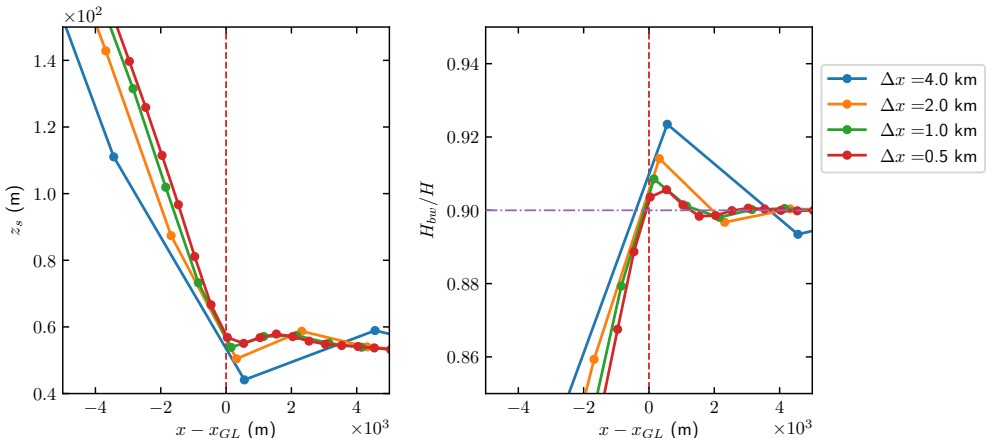

**Figure 7.** Details of the solutions for the advance experiment with $\Delta x = 0.5, 1, 2, 4$ km after 10000 years. The solid dots represent the nodes of the elements and the vertical, red, dashed lines indicate the GL position. *Left panel*: The oscillations in $z_s$ at the ice surface near GL. *Right panel*: The flotation criterion is evaluated by $H_{bw}/H$. The ratio between $\rho/\rho_w$ is drawn in a horizontal, purple, dash-dotted line.

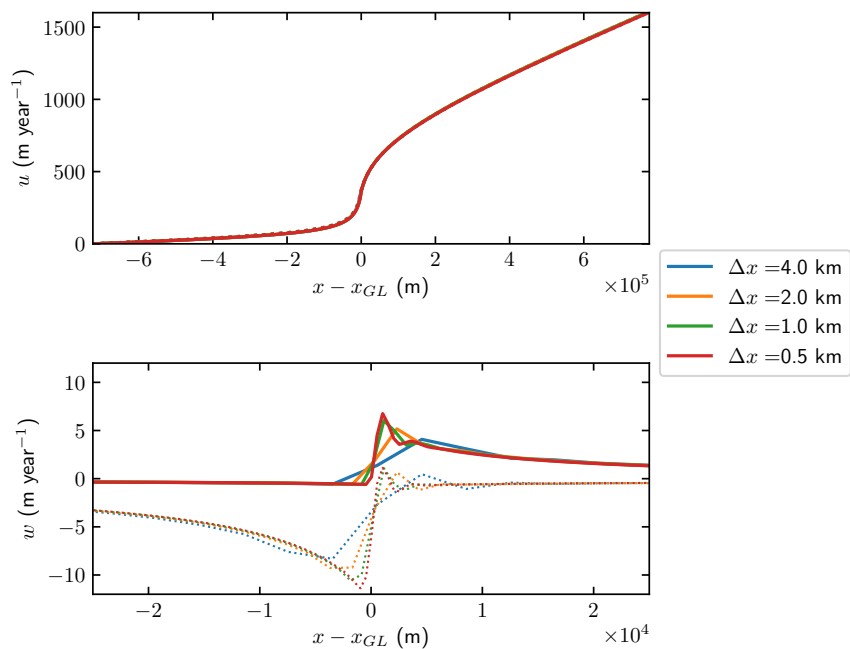

**Figure 8.** The velocities $u$ (upper panel) and $w$ (lower panel) of the ice in the advance experiment with $\Delta x = 0.5, 1, 2, 4$ km after 10000 years. The solutions at the upper surface are the solid curves, and the solutions at the lower surface are the dotted curves. The vertical velocity $w$ is zoomed-in close to the GL with the distance to the mesh dependent GL on the $x$-axis.

## 6 Discussion

Seroussi et al. (2014) describe four different subgrid models (NSEP, SEP1, SEP2 and SEP3) for the friction in SSA and evaluate them in a FEM discretization on a triangulated, planar domain. The flotation criterion is applied at the nodes of the triangles. In the NSEP, an element is floating or not depending on how many of the nodes of that element are floating. In the other three methods, an inner structure in the triangular element is introduced. One part of a triangle is floating and one part is grounded. The amount of friction in a triangle with the GL is determined by the flotation criterion. Either the friction coefficient is reduced, the integration in the element only includes the grounded part, or a higher order polynomial integration (SEP3) is applied. Faster convergence as the mesh is refined is observed for the latter methods compared to the first method. The discretization of the friction in Sect. 4 is similar to the SEP3 method but the FS equations also require a subgrid treatment of the normal velocity condition. In the method for the FS equations in Gagliardini et al. (2016), the GL position is in a node and the friction coefficient is approximated in three different ways. The coefficient is discontinuous at the node in one case (DI in Gagliardini et al. (2016)). Our coefficient is also discontinuous but at the estimated location of the GL between the nodes.

The convergence of the steady state GL position toward the reference solutions in Gagliardini et al. (2016) is observed in the simulations in Fig. 5 and 6. However, as the meshes we used are at least 20 times larger than the 25 m finest resolution in Gagliardini et al. (2016), it has probably not reached the convergence asymptote. At the current resolutions, the discretization introduces a strong mesh effect such as the two different geometrical interpretations in the two cases mentioned in Sect. 4. The subgrid scheme is able to provide a more accurate representation of the GL position and the boundary conditions, but the numerical solution of the velocity field, pressure as well as the two free surfaces are still computed on the coarse mesh, which are the main sources of the numerical errors. Additional uncertainty at the GL is introduced by the approximation of the bedrock geometry, the friction at the GL, and the modeling of the ice/ocean interaction. It is shown in Cheng and Lötstedt (2020) that the solution at the GL is particularly sensitive to variation in the geometry and friction at the ice base.

Our method can be extended to a triangular mesh covering $\Gamma_b$ in the following way (considering linear Lagrange functions). The condition on $\chi$ in Eq. (27) or $\tilde{\chi}$ in Eq. (29) is applied on the edges of each triangle $\mathcal{T}$ in the mesh. If $\chi < 0$ in all three nodes then $\mathcal{T}$ is grounded. If $\chi \geq 0$ in all nodes then $\mathcal{T}$ is floating. The GL passes inside $\mathcal{T}$ if $\chi$ has a different sign in one of the nodes. Then the GL crosses the two edges where $\chi < 0$ in one node and $\chi \geq 0$ in the other node. In this way, a continuous reconstruction of a piecewise linear GL is possible on $\Gamma_b$. The same tests are applied to $\tilde{\chi}$. The FEM approximation is modified in the same manner as in Sect. 4 using step functions in Nitsche's method.

An alternative to a subgrid scheme is to introduce static or dynamic adaptation of the mesh on $\Gamma_b$ with a refinement at the GL as in e.g. Gladstone et al. (2010a); Cornford et al. (2013); Drouet et al. (2013). In general, a fine mesh is needed at the GL and in an area surrounding it. Since the GL moves long distances in simulations of palaeo-ice sheets, the adaptation should be dynamic, permit refinement and coarsening of the mesh varying in time, and be based on some estimate of the numerical error of the method. In shorter time intervals, a static adaptation may be sufficient since the GL will move a shorter distance. Furthermore, shorter time steps are necessary for numerical stability in static and dynamic mesh adaptation schemes. A static adaptation is determined once before the simulation starts. Introducing a time dependent, dynamic mesh with adaptivity into an

existing code requires a substantial coding effort and will increase the computational work considerably compared to a static mesh. Subgrid modeling is easier to implement and the increase in computing time is small. A combination of dynamic mesh adaptation and subgrid discretization may be the ultimate solution. Then the mesh at the GL would be adapted to resolve the variation in the interior of the ice at the GL while the subgrid modeling would handle the discontinuity at the basal boundary.

## 7    Conclusions

A subgrid scheme at the GL has been developed and tested in the SSA model for 2D vertical ice flow in Gladstone et al. (2010b) and in Seroussi et al. (2014), for the friction in the vertically integrated model BISICLES (Cornford et al., 2013) for 2D flow in Cornford et al. (2016), and for the PISM model mixing SIA with SSA in 3D in Feldmann et al. (2014). Here we propose a subgrid scheme for the FS equations for a 2D vertical cross-section of ice, implemented in Elmer/ICE, that can be extended to 3D. The mesh is static and the moving GL position within one element is determined by linear interpolation with an auxiliary function $\chi(\boldsymbol{x})$ or $\tilde{\chi}(\boldsymbol{x})$. Only in that element, the FEM discretization is modified to accommodate the discontinuities in the boundary conditions.

The numerical scheme is applied to the simulation of a 2D vertical ice sheet with an advancing GL and one with a retreating GL. The model setups for the tests are the same as in one of the MISMIP examples (Pattyn et al., 2012) and in Gagliardini et al. (2016). The solution converges smoothly in the neighborhood of the GL when the mesh size is reduced. Comparable results to Gagliardini et al. (2016) are obtained using the subgrid scheme with more than 20 times larger mesh sizes. A larger mesh size also allows a longer time step for the time integration.

*Code availability.* The FS sub-grid model is implemented based on Elmer/ICE Version: 8.3 (Rev: f6bfdc9) with the scripts at http://doi.org/ 10.5281/zenodo.3401478 and http://doi.org/10.5281/zenodo.3401475.

*Author contributions.* GC developed the model code and performed the simulations. GC and PL contributed to the theory of the paper. GC, PL and LvS contributed to the development of the method and the writing of the paper

*Competing interests.* The authors declare that they have no conflict of interest.

*Acknowledgements.* This work has been supported by Nina Kirchner's Formas grant 2017-00665 and the Swedish e-Science initiative eSSENCE. We are grateful to Thomas Zwinger for advise and help in the implementation of the subgrid scheme in Elmer/ICE. The computations were performed on resources provided by the Swedish National Infrastructure for Computing (SNIC) at the PDC Center for High Performance Computing, KTH Royal Institute of Technology. We also thank the anonymous referees for their helpful comments.

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
