# Peer review of "A full Stokes subgrid model for simulation of grounding line migration in ice sheets using Elmer/ICE(v8.3)"

_Geoscientific Model Development, 2019_

## Referee Comment (RC1) · Anonymous Referee #1 · 19 Oct 2019

**Review – Cheng, et al., A full Stokes subgrid model for simulation of grounding line migration in ice sheets using Elmer/ICE (v8.3)**

The manuscript aims to present a numerical scheme to deal with the friction inside elements partly floating in a (full-)Stokes formulation for the marine ice sheet simulation. The formulation and results are carried out in a 2D vertical domain, and possible extension to 3D domain is discussed. Overall the paper is well written, and the figures are well visible. The results are compared to a related study (Glagliardini et al, 2016), where no subgrid scheme is applied: while in the latter the results are achieved using high mesh resolution (50 to 25 m), the current manuscript presents "similar" results using 4000 to 1000 m as mesh resolution.

**General comments:**

- The numerical scheme as well as the equations being solved are presented, although some corrections should be done to make them clearer. Also, it is not clear if there are some iterative steps between the GL position computation and the solver of the FS equations. An overview of the framework involving all the solver processes (each solver step) employed could be useful to make it clearer.
- With the results presented along the manuscript it is hard to analyze the convergence (and consistency) of the subgrid scheme proposed. Also, as mentioned above, the overall explanation on how the entire mathematical problem is solved doesn't help this analyze and could compromise the reproducibility of the results. The results seem promising, however.
- The Introduction section must be improved. The reading is impact by some zigzags. Some simplifications could be done, starting from an overview of the problem and going to the specific problem that is being solved in the manuscript.
- The citation style along the manuscript should be corrected, e.g., in line 89, Hutter (1983) => (Hutter, 1983), and elsewhere.

**Specific comments:**

**2 Ice model**

**2.1 The full Stokes (FS) equations**

- line 89: "2D" => "2D vertical" (and elsewhere)
- line 90: "ice $\Omega$" => "ice domain $\Omega$"
- line 92: the notation here is confused. It should be:

$$\sigma = \tau - p\mathrm{I}$$

where $\tau$ is the deviatoric stress tensor, given by

$$\tau = 2\eta\dot{\varepsilon}$$

where $\dot{\varepsilon}$ is the strain rate tensor, defined as

$$\dot{\varepsilon} = \frac{1}{2}(\nabla \mathbf{u} + \nabla \mathbf{u}^{\mathrm{T}})$$

and $\eta$ is the ice viscosity given by

$$\eta = \frac{1}{2}A^{-\frac{1}{n}}\dot{\varepsilon}_e^{\frac{1-n}{n}}$$

being $\dot{\varepsilon}_e$ defined as

$$\dot{\varepsilon}_e = \sqrt{\frac{1}{2}\mathrm{tr}(\dot{\varepsilon}\dot{\varepsilon}^{\mathrm{T}})}$$

Please, don't use "$\tau$" as strain rate factor, even because it is used in Eq. 10 (and elsewhere) as stress.

**2.2 Boundary conditions**

- line 101: the boundary $\Gamma$ is not represented in Figure 1 (neither $\Gamma_s$ and $\Gamma_b$)
- line 101: "In a 2D case" => "In the 2D vertical case"
- line 102: "y is constant in the figure" => "the ice sheet geometry is constant in y"
- line 102 (and elsewhere): Please, use "ice surface" instead of "upper boundary" or "upper surface", and "ice base" instead of "lower boundary" or "lower surface".
- line 104: The notation here doesn't help. Please, use $\mathbf{f}_s$ only for the forcing applied at the ice surface, $\Gamma_s$. For the floating part of the ice base, $\Gamma_{bf}$, use, for example, $\mathbf{f}_{bf}$.
- line 106: it is good explaining (with few words) the meaning of $\sigma_{nn}$ (the normal component of the force/stress acting on the ice base … ), $\sigma_{nt}$ (the parallel component of the force/stress acting on the ice base …), and $u_t$ (the parallel component of the ice velocity at the ice base …).
- line 108: the same for $u_n$
- line 112: "The GL is located where" => "At the GL,"
- line 112: "In 2D" => "In 2D vertical"
- line 114: "With the ocean surface at $z = 0$, $p_w$ … " => "The ocean surface is at $z = 0$, and $p_w$ …"
- line 115: "gravitation constant" => "gravitational acceleration"

**2.3 The free surface equations**

- line 116: maybe change "free surface equation" to "ice surface and ice base equations"
- line 119 (and elsewhere): "free surface" => "ice surface"
- line 122 (and elsewhere): "lower surface" => "ice base"
- line 122: $z_b$ is negative if below sea level, right?
- line 124: actually, there is just ablation (basal melt) at $\Gamma_{bf}$

**2.4 The solution close to the grounding line**

- line 128: "The 2D" => "The 2D vertical"
- line 130: "the bedrock" => "the grounded part"
- line 130: Which "simple equation" in Schoof 2011? It is not clear (maybe you meant "simple relation")
- line 132: "By adding higher order terms" where? The phrase is not clear
- line 132: "… Archimedes' floatation condition … is not satisfied …" where? In Schoof 2011 analysis? Or considering all the terms in FS? The phrase is not clear
- line 134: does the rapid variation of $w$ appear in Schoof 2011 analysis? The phrase is not clear
- line 136: "height" => "thickness"
- line 136: "length" => "horizontal length"
- line 140: basically, is it assumed that the vertical normal stress ($\sigma_{22}$ or $\sigma_{zz}$) is hydrostatic? Or at least at the ice base (boundary)? See Greve and Blatter, 2009, Eq. 5.59, for example.
- line 141: Does "bedrock" here mean grounded ice or the bedrock, $b(x,z)$? Also, I didn't understand the reference to Eq. 4 and Eq. 6.
- line 143: "Introduce" => "Introducing"
- line 145: "approximate" => "approximating" and "let" => "letting", and "… water surface. Then" => "… water surface, yields"
- line 145: "water surface" => "sea level"
- line 145: What is the reason to "approximate $z_s$ and $z_b$ linearly in $x$"? Both $z_s$ and $z_b$ are linear if the elements are geometrically linear (i.e., the edges of the elements remain straight). Is this the case being referred here? Or are you using this argument to simplify the expression of $\sigma_{zz}$ ($\chi$)?
- line 147: Eq. 12 is a hydrostatic condition to estimate the GL position, right? Why is $H_{bw}$ used instead of the bedrock coordinate, $b(x)$? Using the bedrock instead of $H_{bw}$, the function $\chi$ should be positive for floating ice, and negative for grounded ice. The GL position is found for $\chi = 0$. Using $H_{bw}$, it is not clear how to find the GL position, since $\chi = 0$ for the floating ice, including the GL position, $x_{GL}$. (In fact, this is pointed out in line 272)

**3 Discretization by FEM**

**3.1 The weak form of the FS equations**

- line 155: who is "$n$"? Is "$n$" equal to "$p$" in Chen et al. 2013 notation?
- line 156: please check the definition of the spaces here. It is not clear if "$k$" means k-integrable functions. Also, the space $Q_{k^*}$ and $k^*$ are not well defined. Is it imposed divergence free for $Q_{k^*}$? Or is only the $L^2$ norm required for the pressure space?
- line 161: please, change the notation of the strain rate tensor (as already mentioned above)
- line 163: "size" => "value", and "application" => "physical problem"
- line 163: an observation could be added here, something like: "the sensitivity of the GL positions for different values of $\gamma_0$ is shown in sect. 5"

**3.2 The discretized FS equations**

- line 169: the space $M_{k^*}$ is not defined
- line 170: are the vertical layers equally spaced, for a given $x$?
- line 174: "ice" => "ice sheet"
- line 177: (Eq. 15) the integral limits should something like $s(x_i)$ to $s(x_{GL})$, and $s(x_{GL})$ to $s(x_{i+1})$, where $s$ is the lowest edge (the ice base) of the element $\mathcal{E}_i$ where the GL is located (between $x_i$ and $x_{i+1}$). Maybe a note could be added in the text instead of changing it in the equations (in line 171, for example).
- line 178: this paragraph is not clear. What does it mean "strong formulation"? Also, the basis functions at the lowest edge of the element $\mathcal{E}_i$ (the edge that represents the ice base) are linear, and defined between $s(x_i)$ and $s(x_{i+1})$, right? (here, $s$ is the coordinate along this element's edge). So, even if the integral is split at $s(x_{GL})$, the GL position at the element's edge, there are contribution of the integral on both nodes, $s(x_i)$ and $s(x_{i+1})$, since the basis functions are defined between $s(x_i)$ and $s(x_{i+1})$ (unless additional or modified basis functions are used, similar to X-FEM for example, which it seems it is not the case here). For example, let's take the last integral in Eq. 15, and let's assume (for simplicity) the base is perfectly horizontal between $x_i$ and $x_{i+1}$. The normal vector at the base is $\mathbf{n} = [0\ 1]^T$, and $\mathbf{u} \cdot \mathbf{n} = w$ (vertical velocity). Assuming linear piecewise basis functions ($\phi$) at the nodes $s(x_i)$ and $s(x_{i+1})$, we have $w = w_i\phi_i + w_{i+1}\phi_{i+1}$, and the integral is $\int p_w(w_i\phi_i + w_{i+1}\phi_{i+1})ds$ at that edge, which means that there are contributions of the integral on both nodes, $i$ and $i + 1$. The same happens for the other integral (whose limits are $s(x_i)$ and $s(x_{GL})$). Could you please make that paragraph clearer?
- line 182: "non-linear" => "nonlinear"
- line 199: is "$d_j$" here referred to "$d$", the distance between the base and the bedrock?
- line 199: it is not totally clear how the complementarity problem is solved. During the Newton iterations to solve both $\mathbf{u}$ and $p$, the distance $d$ is kept constant, right (and consequently $p_w$ at the base)? Are $\sigma_{nn}$ and $u_n$ updated at every Newton iteration? So is the complementarity problem (Eqs. 16, 17 and 18) solved at each Newton iteration? How is the GL position defined during this process? By using the function $\chi$ (Eq. 11)? Could you please explain these solver steps?

- line 199: the term "grounded mask" is not used along the text. Could you please explain or change this term? Please, avoid different definitions along the text.

**3.3 Discretization of the advection equations**

- line 203: are you using a complete stabilization scheme, like the Streamline Upwind Petrov-Galerkin (SUPG) scheme, or a scheme based on adding an artificial diffusion, line Artificial Diffusivity or Streamline Upwind?
- line 205: are both advection equations solved together (fully coupled)? Please, could you make it clearer?
- line 209: What does it mean: "The spatial derivative of $z_c$ is approximated by FEM"? Is the derivative of $z_c$ that one provided by the derivative of the basis functions? Or is it applied some gradient recovery method? Please, could you explain it?
- line 215: "Insert" => "Inserting" and "to obtain" => "yields"
- line 217 and Eq. 22: Why is the notation in t different here? In Eq. 19 there is a $t^{n+1}$; here, $t^n$.
- line 218: $z_{bx}$ is not defined in time here (implicit or explicit?)
- line 219: it is not clear, but it seems that both advection equations (or at least Eq. 22) are solved together with the FS equations, Eq. 14. Please, could you explain this? Eq. 19 is semi-implicit, i.e., Eq. 14 is solved first for velocity and pressure, but it is a bit confusing with Eq. 22.
- line 220: "Assume" => "Assuming"
- line 220: "… is small. The timestep …" => "… is small, the timestep …"
- line 223: "Divide" => "Dividing" (and elsewhere along this paragraph and below)
- line 237 (and all paragraph): which exactly scheme is used after all? The semi-implicit for the ice surface and ice base (Eq. 19), or the scheme related to Eq. 22? It is not clear. Could you please add a sequence of the numerical scheme, including the FS equations and the complementarity equations? How is the GL position calculation used here?

**4 Subgrid modeling around grounding line**

- line 242: "Subgrid modeling around grounding line" => "Subgrid scheme around grounding line"
- line 243: what GL parameterization in Seroussi et al., 2014? SEP1, SEP2 or SEP3?
- line 246: please, delete "exact"
- line 245 to 247: "In the Stokes equations, the hydrostatic assumption may not be valid, so the exact GL position can not be determined by simply checking the total thickness of the ice $H$ against the depth below sea level $H_{bw} = -z_b$. " But this assumption is used to deduce Eq. 12.
- line 249: the indicator $\chi$ defined here is different to the indicator defined by Eq. 11 or 12. Also, if $\tau_{22} - p$ is defined by Eq. 10, we have a hydrostatic assumption for $\sigma_{22}$ ($=\sigma_{zz}$, right?). Then, at the end, is a hydrostatic assumption used to define the GL position?
- line 250: "since the slope of the bedrock is small" is it the argument to justify the hydrostatic approximation at the ice base? Could you please explain this phrase?
- line 251: "lower surface" => "ice base"

- line 251: "$z_b > b$" is not the only condition to define the boundary conditions, right? Because, even in the situation of Figure 2 (upper panel), the net force at node $x_{i+1}$ could be pointing outward, i.e., forcing the node to be grounded (e.g., an advance phase of the ice sheet).
- line 253: is the "net force" represented by the "arrows" in Figures 2 and 3?
- line 257: "contact with the bedrock" means $z_b = b$, right?
- line 259: "GL element" is the same element $\mathcal{E}_i$ defined in the line 173, right? Please, try to simplify the number of definitions along the text.
- line 261: The "true position" term here is complicated. Actually, in any fixed mesh with any finite resolution, the "true position" of the GL is not defined. Also, this paragraph seems redundant and could be deleted.
- line 264: Again, the definition of $\chi$. Also, Eq. 11 is a hydrostatic assumption of $\chi$.
- line 266: Note that $\chi(x_i) > 0$ in this case because it is a discrete case ($x_{GL}$ is not perfectly defined). Using Eq. 12, in a continuous case (and with a perfect definition of $x_{GL}$), $\chi \leq 0$, as written in line 148. Maybe a note could be added to avoid confusion.
- line 268: "floating boundary condition" => "floating condition, Eq. (11)" (maybe)
- line 289: The correction made in $\chi$ is only in the pressure water, right? $\sigma_{nn}$ continues as hydrostatic, Eq. 11 or 12, right?
- line 272: why is the bedrock elevation not used in every case (i and ii), if this is the most generic way to solve for $x_{GL}$?
- line 273: "linear functions" => "linearized functions"
- line 274: "As the GL always rests on the bedrock" in theory or ideally, right? Then $p_b(x_{GL})$ is equal to $p_w(x_{GL})$ only when $b(x_{GL}) = z_b(x_{GL})$, what is not the case in the (most) discretization representations. Maybe what you are saying here is that in the solution of $x_{GL}$ using the linear interpolation of $\chi$, $z_b(x)$ should be equal to $b(x)$ at $x_{GL}$, what is not true due to the linear representation of $z_b(x)$, mainly in a coarse mesh resolution (even in a fine resolution, a residual will exist). I think this phrase or even this paragraph could be rewritten. Also, $\sigma_{nn}$ here is that one computed by Eq 10, right? So $\sigma_{nn}$ is hydrostatic, right?
- line 278: what correction do you refer here? Please, could you make this paragraph clearer?
- line 281: Then you increase the number of integration (Gauss) points, right? Similar to SEP3 scheme in Seroussi et al., 2014. What order do you use? Is there any sensitivity in GL positions for different orders?
- line 282: I understand the motivation of "smoothing" the friction coefficient $\beta$ at the GL region, mainly when a Weertman-type friction law is employed (even because $\beta$ "should" be zero at the first floating node, so it seems that 1/2 comes from a linear interpolation of $\beta$ between the last grounded and the first floating nodes). But this "smoothing" effect should already be "captured" by the subgrid scheme you are using (i.e., the basal friction on the GL element is "weighted" by the integration points; it is not a "smoothing" effect in fact, but a reduction of the friction inside the GL element). I am not sure about the effect of multiply $\beta$ by 1/2. Imagine the case where the GL is very close to the first floating node, $x_{i+1}$, and in the next time step, the GL moves to the next (floating) element (defined between $x_{i+1}$ and $x_{i+2}$). So, there is a "jump" in the basal friction

on the previous GL element ($\in [x_i\ x_{i+1}]$), from an almost fully grounded case (with friction coefficient $\beta$ multiplied by 1/2) to a fully grounded case (friction coefficient equal to $\beta$). Have you tested without this "smoothing" (multiplication)?

- line 284: this paragraph is not totally clear. It is not clear how the integral of Eq. 15 is discretized/applied on the GL element. Could you please make this clearer using the same notation of Eq. 15?

- line 290: this paragraph could be a summary of how the equations are solved, but it not totally clear. Are the advection equations solved together with the FS equations, or in a semi-implicit manner? Does the "fixed-point" here refer to the Newton method (line 197) or to a Piccard-like scheme? Also, the high integration order is applied only in the GL element, right?

**5 Results**

- line 296: "and comparison" => "and a comparison" (maybe)
- line 299: "20 vertical extruded layers" equally spaced?
- line 301: I liked this analysis on $\gamma_0$. I suspect the value of $\gamma_0$ should be updated according to the order of magnitude of the matrix coefficient about which the Nitche's term is added (in the stiffness matrix); since it is nonlinear, the order of magnitude of the (stiffness) matrix coefficients change during the simulation (maybe even in each nonlinear iterations). But this is a study to be carried out in the future.
- line 306: "mesh sizes" => "mesh resolutions". The same for line 309.
- line 308: "purple and pink" => "purple and pink, respectively"
- line 308: "We achieve similar GL migration results … with at least 20 times larger mesh sizes." It is an impressive result, and it seems very promising! Some questions arise here. A) It is not expected monotonicity in terms of GL convergence, but it is hard to analyze the convergence looking only at Figure 5. What about plotting also a figure similar to Figure 2 of Gagliardini et al. 2016? B) The GL positions in both phases (advance and retreat) are quite similar using mesh resolution equal to 2 km, but the same is not observed when the resolution is increased (1 km), although the GL positions seem to move (converge?) to the central value (~730 km) with mesh resolution. Is the bedrock description ($b(x)$, given by Eq 16 in Pattyn et al. 2012, right?) at the same resolution of the mesh? If yes, increasing the mesh resolution also increases the bedrock resolution, and we should expect GL convergence to the interval obtained by Gagliardini et al. 2016 (considering that they also increased the bedrock resolution). I think a way to "strengthen" the results is to run the same experiments with one more level of refinement, i.e., with mesh resolution equal to 500 m (if possible, even 250 m). If the bedrock is kept at the coarse resolution (i.e., 4 km in your case here), then the results would not necessarily converge to the Gagliardini interval, and the comparison with their results doesn't seem adequate (although the convergence with mesh resolution should be easier analyzed since the geometry/bedrock is the same for all meshes). Another way to verify the convergence and consistency of the numerical scheme is to run the same kind of experiments using a single-slope bedrock (like MISMIP3D, for example), with different mesh resolutions.

- line 314 (and Fig. 6): it seems that $x_{GL}$ is estimated using $\rho H = \rho_w H_{bw}$. But, it seems that this is not observed in Fig. 6 (right panel) for the GL position presented. Why? In Fig. 6, I expected the red-dashed line (GL) to cross both the purple-dash-dotted line and the green line. Or maybe I am missing something here …
- line 320: "top and bottom" => "surface and base" (and elsewhere in this paragraph)
- line 321: "The horizontal velocities on the two surfaces are similar with negligibly small differences on the floating ice." as expected for the floating ice, right?
- line 323: "… the rapid variation is resolved by the 1 km mesh size." I don't think "resolved" is the right term here. The discontinuity at $x_{GL}$ is not resolved with a continuous space, and the variation of $w$ in $x$ doesn't seem to be a polynomial-type. Maybe "enough approximated" is the term you meant (although "enough" is a matter of discussion, indeed).

**6 Discussion**

- line 326: "floatation criterion" => "hydrostatic floatation criterion"
- line 326: "Depending on how many of the nodes that are floating, the amount of friction in the triangle is determined." Maybe this could be rewritten. Basically, the amount of friction is computed according to the grounded are in partly floating elements. And Seroussi et al., 2014, used different techniques, based on the FEM, to compute this amount of friction.
- line 327: "Also, a higher order polynomial integration over the triangles in FEM allows an inner structure in the triangular element." I didn't understand this phrase; what does it mean "inner structure" here?
- line 330: "… If $\chi \geq 0$ …" $\chi$ is higher than 0 if the water pressure is computed using the bedrock elevation ($b(x)$, as described in line 269). Otherwise, $\chi = 0$ on the floating nodes (as in line 148, unless in the case as shown in Fig. 6).
- line 337: "model inaccuracy" => "numerical error of the model"
- line 340: The subgrid also helps the convergence in comparison to non-subgrid scheme. See for example the comparison between NSEP and SEP1 or SEP2 in Seroussi et al., 2014 (e.g., Fig 2)
- line 340: How does your subgrid implementation compare to the "DI implementation" tested in Gagliardini et al. 2016?

**7 Conclusions**

- line 341: "Subgrid models at the GL" => "Subgrid schemes to model the GL dynamics"
- line 341: "3D flow" actually it is also a SSA-2D flow
- line 342: "3D flow" same here; BISICLES is a "2 1/2 flow", but the grid is 2D (plan x-y)
- line 345: "subgrid model in 2D" => "subgrid scheme in 2D vertical"
- line 346: Note that $\tilde{\chi}$ here is the modified version of $\chi$. Please, try to condensate and simplify the definitions along the text, and try to use just one
- line 348: "method" => "numerical scheme"
- line 348: delete "in 2D"
- line 348: "The data" => "The model setups"

- line 350: "with subgrid modeling" => "using the subgrid scheme"
- line 351: please, delete "Without further knowledge of the basal conditions and detailed models at the GL".
- line 352: "… approximation of the GL position." => "…approximation of the GL position, and accelerates the GL position convergence in comparison to schemes where the GL relies just on element nodes." **Note that this last phrase (suggestion) only makes sense if more numerical tests are performed, helping the convergence analysis of the proposed subgrid scheme.**

As a last suggestion, maybe change "subgrid model" in the title to "subgrid scheme"

**References** (used here)

Gagliardini, O., Brondex, J., Gillet-Chaulet, F., Tavard, L., Peyaud, V., and Durand, G.: Impact of mesh resolution for MISMIP and MIS- MIP3d experiments using Elmer/ICE, The Cryosphere, 10, 307–312, 2016.

Seroussi, H., Morlighem, M., Larour, E., Rignot, E., and Khazendar, A.: Hydrostatic grounding line parameterization in ice sheet models, Cryosphere, 8, 2075–2087, 2014.

Schoof, C.: Marine ice sheet dynamics. Part 2. A Stokes flow contact problem, J. Fluid Mech., 679, 122–155, 2011.

Hutter, K.: Theoretical Glaciology, D. Reidel Publishing Company, Terra Scientific Publishing Company, Dordrecht, 1983.

Chen, Q., Gunzburger, M., and Perego, M.: Well-posedness results for a nonlinear Stokes problem arising in glaciology, SIAM Journal on Mathematical Analysis, 45, 2710–2733, 2013.

Greve, R., Blatter, H. Dynamics of Ice Sheets and Glaciers. Advances in Geophysical and Environmental Mechanics and Mathematics. Springer-Verlag Berlin Heidelberg, Berlin, Germany, 1th edition, 2009.

---

## Referee Comment (RC2) · Anonymous Referee #2 · 29 Oct 2019

This paper tries to implement the subgrid scheme of grounding line (GL) movements in Elmer/Ice and test it with the 2D MISMIP benchmark. The "full" Stokes model is computational intense, especially for solving marine ice sheet problems where very fine mesh resolution is usually needed around GL to accurately capture the movement of GL. Thus, a subgrid scheme study like this paper is certainly valuable. However, the current version of this manuscript is probably not ready yet for a consideration of publication, due to the following reasons:

1. The authors use a hydrostatic (first-order) approximation to determine whether a node is floating or grounded, which looks disappointing for a "full" Stokes model. In

Elmer/Ice, to solve the contact problem of GL dynamics, the normal stress (nodal force) was actually used in previous studies (e.g., Durand et al., 2009). It's surprising that this paper doesn't use that, which, from my point of view, is not a Stokes solution.

2. For testing the capability of the subgrid scheme, the authors should at least do the MISMIP3d experiments, in order to see how the GL move in y. The MISMIP benchmark is good, but I don't understand why not trying the MISMIP3d since it's been out there for a couple of years. The authors have some discussions of extending the 2d implementation to 3d, but perhaps it's better to just test it.

3. The writings, particularly the introduction section, still needs improvement.

Some apparent technical comments:

- Citation style throughout the whole paper needs to be corrected

- In the title, there should be a space right after Elmer/ICE, or just remove (v8.3)

- Line 13: change it to "...an indicator of ice sheet advances or retreats"

- Line 15: "on West Antarctica" to "in West Antarctica"

- Line 18: In theory the Stokes model is the most accurate, but in reality it might not the one that shows the best match to observations. So please make an explicit and correct statement.

- Line 25: It is the longitudinal stress gradient, not longitudinal stress, that controls the flow of ice shelf.

- Line 121: change "net surface accumulation/ablation" to "surface mass balance"

- Line 124: change "net accumulation/ablation at the lower surface" to "basal mass balance"

- Line 134: change "short interval" to "short distance interval"

- Line 270-273: For case i, how come the GL position is at the floating part? It may

look reasonable numerically, but it's totally not physical.

References:

Durand, G., Gagliardini, O., De Fleurian, B., Zwinger, T. and Le Meur, E., 2009. Marine ice sheet dynamics: Hysteresis and neutral equilibrium. Journal of Geophysical Research: Earth Surface, 114(F3).

―――――――――――――――――――――――

---

## Referee Comment (RC3) · Anonymous Referee #3 · 4 Nov 2019

This paper presents a subgrid interpolation across the grounding line for the Stokes equation. This interpolation is based on the stress balance at the base of the ice, $\chi = p_w + \sigma_{nn}$ with $\sigma_{nn}$ the normal deviatoric stress at the ice base, and $p_w$ the water pressure. If the last grounded element is at node $i$, and the first floating element is at $i + 1$, then the position $x_{GL}$ of the grounding line is determined as the first point where $\chi$ goes to zero. This position is then used in the evaluation of the weak form of the basal boundary condition.

Unfortunately, the incorrect citation style and at times awkward writing of the paper (in particular the introduction) distract from its contents. These need to be corrected

before publication can be considered. Scientifically, the approach is a logical first step for interpolation of the basal boundary condition across the grounding line, and the same approach has been used in Seroussi et al. (2004), though not for a Full Stokes model. That said, I think that the analysis of the results could be improved significantly by

1. comparison with other interpolation schemes in 1 horizontal dimension, and

2. extension to 2 horizontal dimensions. The latter case is briefly discussed in lines 229-333, but I think an implementation would show whether this approach in indeed able to deal with complex grounding line geometries.

These kinds of comparisons are standard for the study of numerical grounding line schemes (see e.g., Seroussi et al., 2004, Feldmann et al., 2014).

For point 1 above, I am specifically interested in seeing a comparison to an interpolation of the basal shear stress constant $\beta$ between the last grounded and the first floating point, i.e., if one would multiply $\beta$ with $(x_{GL} - x_i)/(x_{i+1} - x_i)$, how would the results differ? This is the interpolation scheme traditionally used in depth-integrated models (e.g., Feldmann et al., 2014, Pattyn et al., 2006) and also introduced as SEP1 in Seroussi et al. (2014). I am wondering whether such an interpolation alone would already provide the observed improvement in the numerical performance, as suggested from my reading of Seroussi et al. (2014). I am also sceptical about the effect of setting $\beta$ to $\beta/2$ in the interpolated cell, as suggested in lines 282-283? This introduces an additional interpolation which is similar to the interpolations used in models with a structured grid, and has no physical basis in the presented scheme as the interpolation is already done by splitting up the integral for the boundary condition.

Moreover, it wasn't completely clear to me which parts of the FEM implementation in Elmer/Ice were actually altered. For example, my impression of the time-stepping

scheme is that it is basically the same as in Durand et al. (2009a), in which case section 3.3. is unnecessary.

Other comments (kept short as I think the entire paper needs to be rewritten):

- Better use $\dot{\varepsilon}$ for the strain rate, $\tau$ is usually used for a stress tensor

- Equations (7) and (8) are the kinematic boundary conditions, and should be referred to as such
* * *

---

## Author Comment (AC1) · 10 Dec 2019

article [english]babel amsmath amssymb color

**Response to Anonymous Referee #1**

December 10, 2019

The manuscript aims to present a numerical scheme to deal with the friction inside elements partly floating in a (full-)Stokes formulation for the marine ice sheet simulation. The formulation and results are carried out in a 2D vertical domain, and possible extension to 3D domain is discussed. Overall the paper is well written, and the figures are well visible. The results are compared to a related study (Gagliardini et al, 2016), where no subgrid scheme is applied: while in the latter the results are achieved using high mesh resolution (50 to 25 m), the current manuscript presents "similar" results using 4000 to 1000 m as mesh resolution.

**Response:** We really appreciate the anonymous referee for the detailed reviewing and providing so many helpful comments to improve the quality of this paper. The line numbers below refer to the numbers in the original version of the paper.

**1 General comments**

1. The numerical scheme as well as the equations being solved are presented, although some corrections should be done to make them clearer. Also, it is not clear if there are some iterative steps between the GL position computation and

the solver of the FS equations. An overview of the framework involving all the solver processes (each solver step) employed could be useful to make it clearer.
**Response:** Thanks for the comments. The whole Section 4 has been rewritten to make it more clear. Three algorithms are given explicitly.

2. With the results presented along the manuscript it is hard to analyze the convergence (and consistency) of the subgrid scheme proposed. Also, as mentioned above, the overall explanation on how the entire mathematical problem is solved doesn't help this analyze and could compromise the reproducibility of the results. The results seem promising, however.
**Response:** We agree that the convergence is hard to analyze. The reason is that the mesh size we use here is too large so the solution is far from the asymptotic mesh convergence for the solution with a singularity on the boundary. We only show that the subgrid solutions are all in the range of the finest resolution.

3. The Introduction section must be improved. The reading is impact by some zigzags. Some simplifications could be done, starting from an overview of the problem and going to the specific problem that is being solved in the manuscript.
**Response:** The Introduction has been shortened by removing parts of the overview and concentrating on earlier results more relevant to our own contribution, and other parts have been rewritten.

4. The citation style along the manuscript should be corrected, e.g., in line 89, Hutter (1983) => (Hutter, 1983), and elsewhere
**Response:** The correction has been made.
[Figure]

**2 Specific comments**

2 Ice model

2.1 The full Stokes (FS) equations

- line 89: "2D" => "2D vertical" (and elsewhere)
  **Response:** The correction has been made.

- line 90: "ice $\Omega$" => "ice domain $\Omega$"
  **Response:** The correction has been made.

- line 92: the notation here is confused. It should be:

$$\sigma = \tau - pI$$

where $\tau$ is the deviatoric stress tensor, given by

$$\tau = 2\eta\dot{\epsilon}$$

where $\dot{\epsilon}$ is the strain rate tensor, defined as

$$\dot{\epsilon} = \frac{1}{2}(\nabla\mathbf{u} + \nabla\mathbf{u}^T)$$

and $\eta$ is the ice viscosity given by

$$\eta = \frac{1}{2}A^{-\frac{1}{n}}\dot{\epsilon}_e^{\frac{1-n}{n}}$$

being $\dot{\epsilon}_e$ defined as

$$\dot{\epsilon}_e = \sqrt{\frac{1}{2}tr(\dot{\epsilon}\dot{\epsilon}^T)}$$

[Figure]

Please, don't use $\tau$ as strain rate factor, even because it is used in Eq. 10 (and elsewhere) as stress.
**Response:** The correction has been made.

2.2 Boundary conditions

- line 101: the boundary $\Gamma$ is not represented in Figure 1 (neither $\Gamma_s$ and $\Gamma_b$)
  **Response:** $\Gamma_s,\Gamma_{bf}$ and $\Gamma_{bg}$ are added in Figure 1, $\Gamma_b = \Gamma_{bg} \cup \Gamma_{bf}$. $\Gamma$ is the boundary of the whole domain $\Omega$.

- line 101: "In a 2D case" $=>$ "In the 2D vertical case"
  **Response:** The correction has been made.

- line 102: "$y$ is constant in the figure" $=>$ "the ice sheet geometry is constant in $y$"
  **Response:** The correction has been made.

- line 102 (and elsewhere): Please, use "ice surface" instead of "upper boundary" or "upper surface", and "ice base" instead of "lower boundary" or "lower surface".
  **Response:** The correction has been made.

- line 104: The notation here doesn't help. Please, use $f_s$ only for the forcing applied at the ice surface, $\Gamma_s$. For the floating part of the ice base, $\Gamma_{bf}$, use, for example, $f_{bf}$.
  **Response:** The correction has been made.

- line 106: it is good explaining (with few words) the meaning of $\sigma_{nn}$ (the normal component of the force/stress acting on the ice base . . . ), $\sigma_{nt}$ (the parallel component of the force/stress acting on the ice base . . .), and $u_t$ (the parallel component of the ice velocity at the ice base . . .).
  **Response:** The correction has been made.

- line 108: the same for $u_n$
  **Response:** The correction has been made.

- line 112: "The GL is located where" $=>$ "At the GL,"
  **Response:** The correction has been made.

- line 112: "In 2D" $=>$ "In 2D vertical"
  **Response:** The correction has been made.

- line 114: "With the ocean surface at $z = 0$, $p_w$ ... " $=>$ "The ocean surface is at $z = 0$, and $p_w$ ..."
  **Response:** The correction has been made.

- line 115: "gravitation constant" $=>$ "gravitational acceleration"
  **Response:** The correction has been made.

2.3 The free surface equations

- line 116: maybe change "free surface equation" to "ice surface and ice base equations"
  **Response:** This term "free surface equation" is used by Elmer/ICE and most of the papers related to it. We would like to keep it as it was.

- line 119 (and elsewhere): "free surface" $=>$ "ice surface"
  **Response:** The correction has been made.

- line 122 (and elsewhere): "lower surface" $=>$ "ice base"
  **Response:** The correction has been made.

- line 122: $z_b$ is negative if below sea level, right?
  **Response:** Yes, it is negative below sea level as the coordinate system indicated in Figure 1.

[Figure]

- line 124: actually, there is just ablation (basal melt) at $\Gamma_{bf}$
  **Response:** The correction has been made.

2.4 The solution close to the grounding line

- line 128: "The 2D" $=>$ "The 2D vertical"
  **Response:** The correction has been made.

- line 130: "the bedrock" $=>$ "the grounded part"
  **Response:** The correction has been made.

- line 130: Which "simple equation" in Schoof 2011? It is not clear (maybe you meant "simple relation")
  **Response:** Changed to 'simple relation'

- line 132: "By adding higher order terms" where? The phrase is not clear
  **Response:** The correction has been made.

- line 132: ". . . Archimedes' floatation condition . . . is not satisfied . . ." where? In Schoof 2011 analysis? Or considering all the terms in FS? The phrase is not clear
  **Response:** The sentence is rewritten

- line 134: does the rapid variation of $w$ appear in Schoof 2011 analysis? The phrase is not clear
  **Response:** 'in the analysis' is added

- line 136: "height" $=>$ "thickness"
  **Response:** The correction has been made.

- line 136: "length" $=>$ "horizontal length"
  **Response:** The correction has been made.

- line 140: basically, is it assumed that the vertical normal stress ($\sigma_{22}$ or $\sigma_{zz}$) is hydrostatic? Or at least at the ice base (boundary)? See Greve and Blatter, 2009, Eq. 5.59, for example.
  **Response:** No, it is not an assumption. It follows from equations (4.25) and (4.26) in Schoof's paper.

- line 141: Does "bedrock" here mean grounded ice or the bedrock, $b(x,z)$? Also, I didn't understand the reference to Eq. 4 and Eq. 6.
  **Response:** The discussion is rephrased

- line 143: "Introduce" $=>$ "Introducing"
  **Response:** The correction has been made.

- line 145: "approximate" $=>$ "approximating" and "let" $=>$ "letting", and "... water surface. Then "$=>$ "... water surface, yields"
  **Response:** The correction has been made.

- line 145: "water surface" $=>$ "sea level"
  **Response:** The correction has been made.

- line 145: What is the reason to "approximate $z_b$ and $z_s$ linearly in $x$"? Both $z_b$ and $z_s$ are linear if the elements are geometrically linear (i.e., the edges of the elements remain straight). Is this the case being referred here? Or are you using this argument to simplify the expression of $\sigma_{zz}(\chi)$?
  **Response:** This assumption is not necessary here and has been removed

- line 147: Eq. 12 is a hydrostatic condition to estimate the GL position, right? Why is $H_{bw}$ used instead of the bedrock coordinate, b(x)? Using the bedrock instead of $H_{bw}$, the function $\chi$ should be positive for floating ice, and negative for grounded ice. The GL position is found for $\chi = 0$. Using $H_{bw}$, it is not clear how to find the GL position, since $\chi = 0$ for the floating ice, including the GL position,

$x_{GL}$. (In fact, this is pointed out in line 272)

**Response:** The reason is that $\chi < 0$ for $x < x_{GL}$ and $\chi = 0$ for $x > x_{GL}$. It is explained that the formula here is an approximation of the condition we use. How the GL is found by linear approximation is described in Section 4.

3 Discretization by FEM

3.1 The weak form of the FS equations

- line 155: who is "$n$"? Is "$n$" equal to "$p$" in Chen et al. 2013 notation?
  **Response:** It is $n$ in Glen's flow law.

- line 156: please check the definition of the spaces here. It is not clear if "$k$" means $k-$integrable functions. Also, the space $Q_{k^*}$ and $k^*$ are not well defined. Is it imposed divergence free for $Q_{k^*}$? Or is only the $L^2$ norm required for the pressure space?
  **Response:** This is standard notation. We have added one more reference where the same notation is used. The space $Q_{k^*}$ is not divergence free and the pressure is in $L^{k^*}$.

- line 161: please, change the notation of the strain rate tensor (as already mentioned above)
  **Response:** The correction has been made.

- line 163: "size" $=>$ "value", and "application" $=>$ "physical problem"
  **Response:** The correction has been made.

- line 163: an observation could be added here, something like: "the sensitivity of the GL positions for different values of $\gamma_0$ is shown in sect. 5"
  **Response:** The correction has been made.

**3.2 The discretized FS equations**

- line 169: the space $M_{k^*}$ is not defined
  **Response:** The correction has been made.

- line 170: are the vertical layers equally spaced, for a given $x$?
  **Response:** The correction has been made.

- line 174: "ice" $=>$ "ice sheet"
  **Response:** The correction has been made.

- line 177: (Eq. 15) the integral limits should something like $s(x_i)$ to $s(x_{GL})$, and $s(x_{GL})$ to $s(x_{i+1})$, where $s$ is the lowest edge (the ice base) of the element $\mathcal{E}_i$ where the GL is located (between $x_i$ and $x_{i+1}$. Maybe a note could be added in the text instead of changing it in the equations (in line 171, for example).
  **Response:** The notation is improved now.

- line 178: this paragraph is not clear. What does it mean "strong formulation"? Also, the basis functions at the lowest edge of the element $\mathcal{E}_i$ (the edge that represents the ice base) are linear, and defined between $s(x_i)$ to $s(x_{i+1})$, right? (here, $s$ is the coordinate along this element's edge). So, even if the integral is split at $s(x_{GL})$, the GL position at the element's edge, there are contribution of the integral on both nodes, $s(x_i)$ to $s(x_{i+1})$, since the basis functions are defined between $s(x_i)$ to $s(x_{i+1})$ (unless additional or modified basis functions are used, similar to XFEM for example, which it seems it is not the case here). For example, let's take the last integral in Eq. 15, and let's assume (for simplicity) the base is perfectly horizontal between $x_i$ and $x_{i+1}$. The normal vector at the base is $n = [0\,1]^T$, and $\mathbf{u} \cdot n = w$ (vertical velocity). Assuming linear piecewise basis functions ($\phi$) at the nodes $s(x_i)$ to $s(x_{i+1})$, we have $w = w_i \phi_i + w_{i+1} \phi_{i+1}$, and the integral is $\int p_w (w_i \phi_i + w_{i+1} \phi_{i+1})$ at that edge, which means that there are contributions of the integral on both nodes, $i$ and $i+1$. The same happens for

the other integral (whose limits are $s(x_i)$ to $s(x_{GL})$. Could you please make that paragraph clearer?

**Response:** Strong and weak formulations are explained in a better way. The integration intervals are better explained (see the above comment). The basis functions are bilinear on rectangular elements in the $(x, z)$ coordinates but behave essentially as linear ones on the boundary. The base $(x, b(x))$ is represented by a piecewise linear approximation between the nodes. Then it is natural to use a linear function $\tilde{\chi}$ to find the GL by linear interpolation.

- line 182: "non-linear" $=>$ "nonlinear"
  **Response:** The correction has been made.

- line 199: is "$d_j$" here referred to "$d$", the distance between the base and the bedrock?
  **Response:** Yes, $d_j = d(x_j)$ is from the numerical solution.

- line 199: it is not totally clear how the complementarity problem is solved. During the Newton iterations to solve both $\mathbf{u}$ and $p$, the distance $d$ is kept constant, right (and consequently $p_w$ at the base)? Are $\sigma_{nn}$ and $\mathbf{u}_n$ updated at every Newton iteration? So is the complementarity problem(Eqs. 16, 17 and 18) solved at each Newton iteration? How is the GL position defined during this process? By using the function $\chi$ (Eq. 11)? Could you please explain these solver steps?
  **Response:** The solution procedure is added explicitly in Algorithm 1 and 3.

- line 199: the term "grounded mask" is not used along the text. Could you please explain or change this term? Please, avoid different definitions along the text.
  **Response:** The term 'grounded mask' is removed. It is replaced by the two step functions in Sect. 4 and explained with more details in text and figures there.

[Figure]

3.3 Discretization of the advection equations

- line 203: are you using a complete stabilization scheme, like the Streamline Up-wind PetrovGalerkin (SUPG) scheme, or a scheme based on adding an artificial diffusion, line Artificial Diffusivity or Streamline Upwind?
  **Response:** 'artificial diffusion' was used. A sentence is added in the manuscript.

- line 205: are both advection equations solved together (fully coupled)? Please, could you make it clearer?
  **Response:** Algorithm 2 is added to show the time scheme used.

- line 209: What does it mean: "The spatial derivative of $z_c$ is approximated by FEM"? Is the derivative of $z_c$ that one provided by the derivative of the basis functions? Or is it applied some gradient recovery method? Please, could you explain it?
  **Response:** The spatial discretization is explained in the first paragraph of this section.

- line 215: "Insert" $=>$ "Inserting" and "to obtain" $=>$ "yields"
  **Response:** The correction has been made.

- line 217 and Eq. 22: Why is the notation in $t$ different here? In Eq. 19 there is a $t^{n+1}$; here, $t^n$.
  **Response:** The time schemes are all changed to $t^{n+1}$.

- line 218: $z_{bx}$ is not defined in time here (implicit or explicit?)
  **Response:** The time level of $z_{bx}$ is added in Eq. (22), which is an implicit scheme.

- line 219: it is not clear, but it seems that both advection equations (or at least Eq. 22) are solved together with the FS equations, Eq. 14. Please, could you explain this? Eq. 19 is semi-implicit, i.e., Eq. 14 is solved first for velocity and pressure,

but it is a bit confusing with Eq. 22.
**Response:** The time scheme is added as in Algorithm 2.

- line 220: "Assume" => "Assuming"
  **Response:** The correction has been made.

- line 220: "... is small. The timestep ..." => "... is small, the timestep ..."
  **Response:** The correction has been made.

- line 223: "Divide" => "Dividing" (and elsewhere along this paragraph and below)
  **Response:** The correction has been made.

- line 237 (and all paragraph): which exactly scheme is used after all? The semi-implicit for the ice surface and ice base (Eq. 19), or the scheme related to Eq. 22? It is not clear. Could you please add a sequence of the numerical scheme, including the FS equations and the complementarity equations? How is the GL position calculation used here?
  **Response:** The time scheme is added as in Algorithm 2 and the procedure of solving FS is added as in Algorithm 1.

4 Subgrid modeling around grounding line

**Response:** The whole Section 4 has been rewritten to present the subgrid model in a more clear way.

- line 242: "Subgrid modeling around grounding line" => "Subgrid scheme around grounding line"
  **Response:** The correction has been made.

- line 243: what GL parameterization in Seroussi et al., 2014? SEP1, SEP2 or SEP3?
  **Response:** It follows the idea SEP3.

- line 246: please, delete "exact"
  **Response:** The correction has been made.

- line 245 to 247: "In the Stokes equations, the hydrostatic assumption may not be valid, so the exact GL position can not be determined by simply checking the total thickness of the ice against the depth below sea level $H_{bw} = -z_b$." But this assumption is used to deduce Eq. 12.
  **Response:** We have rephrased that. Eq. 12 is an approximation of the estimated GL position. That is the inspiration of the idea to use linear interpolation in this section to determine the GL position.

- line 249: the indicator $\chi$ defined here is different to the indicator defined by Eq. 11 or 12. Also, if $\tau_{22} - p$ is defined by Eq. 10, we have a hydrostatic assumption for $\sigma_{22}$ ($= \sigma_{zz}$, right?). Then, at the end, is a hydrostatic assumption used to define the GL position?
  **Response:** We have redefined the variable in Eq. 11 and 12 to $\chi_a$, and used $\chi(x) = \sigma_{\mathbf{nn}}(x) + p_w(x)$ here.

- line 250: "since the slope of the bedrock is small" is it the argument to justify the hydrostatic approximation at the ice base? Could you please explain this phrase?
  **Response:** Yes, it is. That is only used in $\chi_a$.

- line 251: "lower surface" $=>$ "ice base"
  **Response:** The correction has been made.

- line 251: "$z_b > b$" is not the only condition to define the boundary conditions, right? Because, even in the situation of Figure 2 (upper panel), the net force at node $x_{i+1}$ could be pointing outward, i.e., forcing the node to be grounded (e.g., an advance phase of the ice sheet).
  **Response:** That is only the geometrical condition. To define the boundary, it

has to be combined with the stress balance conditions in Eq(5) and/or (6). This sentence is rephrased.

- line 253: is the "net force" represented by the "arrows" in Figures 2 and 3?
  **Response:** Yes, descriptions have been added in the figures.

- line 257: "contact with the bedrock" means $z_b = b$, right?
  **Response:** Thanks, $z_b = b$ is added.

- line 259: "GL element" is the same element $\mathcal{E}_i$ defined in the line 173, right? Please, try to simplify the number of definitions along the text.
  **Response:** $\mathcal{E}_i$ is the GL element if we use the $i$ index as in Fig. 2 and 3. We change this to be a more general notation as $\mathcal{E}_j$.

- line 261: The "true position" term here is complicated. Actually, in any fixed mesh with any finite resolution, the "true position" of the GL is not defined. Also, this paragraph seems redundant and could be deleted.
  **Response:** We have changed this term to the 'analytical solution' and rewritten the description for it.

- line 264: Again, the definition of $\chi$. Also, Eq. 11 is a hydrostatic assumption of $\chi$.
  **Response:** The correction has been made.

- line 266: Note that $\chi(x_i) > 0$ in this case because it is a discrete case ($x_{GL}$ is not perfectly defined). Using Eq. 12, in a continuous case (and with a perfect definition of $x_{GL}$), $\chi(x_i) \leq 0$, as written in line 148. Maybe a note could be added to avoid confusion.
  **Response:** Thanks. This part is also rephrased.

- line 268: "floating boundary condition" $=>$ "floating condition, Eq. (11)" (maybe)
  **Response:** The correction has been made.

[Figure]

- line 289: The correction made in $\chi$ is only in the pressure water, right? $\sigma_{nn}$ continues as hydrostatic, Eq. 11 or 12, right?
  **Response:** Yes, we have changed the way to present case i and ii, to be more clear. The correction is made on the water pressure, but $\sigma_{nn}$ is from the numerical solution of FS, not Eq.11 or 12.

- line 272: why is the bedrock elevation not used in every case (i and ii), if this is the most generic way to solve for $x_{GL}$?
  **Response:** This part is also rephrased to eliminate confusion. New Figures 2 and 3 for case i and ii are presented.

- line 273: "linear functions" $=>$ "linearized functions"
  **Response:** The correction has been made.

- line 274: "As the GL always rests on the bedrock" in theory or ideally, right? Then $p_b(x_{GL})$ is equal to $p_w(x_{GL})$ only when $b(x_{GL}) = z_b(x_{GL})$, what is not the case in the (most) discretization representations. Maybe what you are saying here is that in the solution of $x_{GL}$ using the linear interpolation of $z_b(x)$ should be equal to $b(x)$ at $x_{GL}$, what is not true due to the linear representation of $z_b(x)$, mainly in a coarse mesh resolution (even in a fine resolution, a residual will exist). I think this phrase or even this paragraph could be rewritten. Also, $\sigma_{nn}$ here is that one computed by Eq 10, right? So $\sigma_{nn}$ is hydrostatic, right?
  **Response:** Thanks for the comments. This whole paragraph has been rewritten accordingly and also the way to present the two cases.

- line 278: what correction do you refer here? Please, could you make this paragraph clearer?
  **Response:** The correction has been made.

- line 281: Then you increase the number of integration (Gauss) points, right? Similar to SEP3 scheme in Seroussi et al., 2014. What order do you use? Is

there any sensitivity in GL positions for different orders?
**Response:** Since Elmer/ICE only support up to 13th order, and according to Seroussi et al., 2014, we only tried with higher than 10th order. We did not observe any differences at between 10th to 13th.

- line 282: I understand the motivation of "smoothing" the friction coefficient $\beta$ at the GL region, mainly when a Weertman-type friction law is employed (even because $\beta$ "should" be zero at the first floating node, so it seems that 1/2 comes from a linear interpolation of $\beta$ between the last grounded and the first floating nodes). But this "smoothing" effect should already be "captured" by the subgrid scheme you are using (i.e., the basal friction on the GL element is "weighted" by the integration points; it is not a "smoothing" effect in fact, but a reduction of the friction inside the GL element). I am not sure about the effect of multiply $\beta$ by 1/2. Imagine the case where the GL is very close to the first floating node, $x_{i+1}$, and in the next time step, the GL moves to the next (floating) element (defined between $x_{i+1}$ and $x_{i+2}$). So, there is a "jump" in the basal friction on the previous GL element ($\in [x_i, x_{i+1}]$), from an almost fully grounded case (with friction coefficient $\beta$ multiplied by 1/2) to a fully grounded case (friction coefficient equal to $\beta$). Have you tested without this "smoothing" (multiplication)?
**Response:** Thanks for the comments. We fully agree. However, there was a mistake in describing the smoothing of $\beta$ at GL element. We have added two new panels in the two cases to show how the Nitsche's method and smoothing of $\beta$ is done at GL, according to the estimated GL position.

- line 284: this paragraph is not totally clear. It is not clear how the integral of Eq. 15 is discretized/applied on the GL element. Could you please make this clearer using the same notation of Eq. 15?
**Response:** A paragraph and Eq. 27 is added to show the implementation. Algorithm 3 is added to show how the subgrid model is implemented.

- line 290: this paragraph could be a summary of how the equations are solved, but it not totally clear. Are the advection equations solved together with the FS equations, or in a semi-implicit manner? Does the "fixed-point" here refer to the Newton method (line 197) or to a Piccard-like scheme? Also, the high integration order is applied only in the GL element, right?
  **Response:** The paragraph has be rewritten with all these technical details.

5 Results

- line 296: "and comparison" $=>$ "and a comparison" (maybe)
  **Response:** The correction has been made.

- line 299: "20 vertical extruded layers" equally spaced?
  **Response:** Yes, they are. This is explained in a better way now.

- line 301: I liked this analysis on $\gamma_0$. I suspect the value of $\gamma_0$ should be updated according to the order of magnitude of the matrix coefficient about which the Nitsche's term is added (in the stiffness matrix); since it is nonlinear, the order of magnitude of the (stiffness) matrix coefficients change during the simulation (maybe even in each nonlinear iterations). But this is a study to be carried out in the future.
  **Response:** Thanks for this comment. This could be interesting to investigate in the future. The choice of the fixed $\gamma_0$ is based on the results from the the two papers using Nitsche's method (; ).

- line 306: "mesh sizes" $=>$ "mesh resolutions". The same for line 309.
  **Response:** The correction has been made.

- line 308: "purple and pink" $=>$ "purple and pink, respectively"
  **Response:** The correction has been made.

- line 308: "We achieve similar GL migration results . . . with at least 20 times larger mesh sizes." It is an impressive result, and it seems very promising! Some questions arise here. A) It is not expected monotonicity in terms of GL convergence, but it is hard to analyze the convergence looking only at Figure 5. What about plotting also a figure similar to Figure 2 of Gagliardini et al. 2016? B) The GL positions in both phases (advance and retreat) are quite similar using mesh resolution equal to 2 km, but the same is not observed when the resolution is increased (1 km), although the GL positions seem to move (converge?) to the central value ( 730 km) with mesh resolution. Is the bedrock description ($b(x)$, given by Eq 16 in Pattyn et al. 2012, right?) at the same resolution of the mesh? If yes, increasing the mesh resolution also increases the bedrock resolution, and we should expect GL convergence to the interval obtained by Gagliardini et al. 2016 (considering that they also increased the bedrock resolution). I think a way to "strengthen" the results is to run the same experiments with one more level of refinement, i.e., with mesh resolution equal to 500 m (if possible, even 250 m). If the bedrock is kept at the coarse resolution (i.e., 4 km in your case here), then the results would not necessarily converge to the Gagliardini interval, and the comparison with their results doesn't seem adequate (although the convergence with mesh resolution should be easier analyzed since the geometry/bedrock is the same for all meshes). Another way to verify the convergence and consistency of the numerical scheme is to run the same kind of experiments using a single-slope bedrock (like MISMIP3D, for example), with different mesh resolutions.

**Response:** We expect the convergence of our method to behave regularly and asymptotically correct when $\Delta x \to 0$ in the same way as you see in Gagliardini et al. 2016. For very small $\Delta x$ the position of the GL inside the GL element does not matter. The reason why we do not observe the asymptotic behavior is that we are not close to the asymptotic regime for $\Delta x$ because we wish our method to allow for larger mesh sizes and still have decent solutions around the GL. For the interior, $\Delta x = 1 km$ may be sufficient (), but in combination with the discontinuous

boundary condition it is insufficient without a special treatment of the GL element.

- line 314 (and Fig. 6): it seems that $x_{GL}$ is estimated using $\rho H = \rho_w H_{bw}$. But, it seems that this is not observed in Fig. 6 (right panel) for the GL position presented. Why? In Fig. 6, I expected the red-dashed line (GL) to cross both the purple-dash-dotted line and the green line. Or maybe I am missing something here . . .
  **Response:** The position of GL is not estimated by the $\rho H = \rho_w H_{bw}$. Instead, the paper solves $\tilde{\chi}(\mathbf{x}_{GL}) = 0$ to find it. If the flotation condition is valid at the GL then the three lines should cross at the same point, but the GL position is estimated by the solution of the FS equation and this may not satisfy the flotation condition exactly. The first term in the small parameter expansion satisfies eq. (9).

- line 320: "top and bottom" $=>$ "surface and base" (and elsewhere in this paragraph)
  **Response:** The correction has been made.

- line 321: "The horizontal velocities on the two surfaces are similar with negligibly small differences on the floating ice." as expected for the floating ice, right?
  **Response:** Yes, this is expected.

- line 323: ". . . the rapid variation is resolved by the 1 km mesh size." I don't think "resolved" is the right term here. The discontinuity at $x_{GL}$ is not resolved with a continuous space, and the variation of $w$ in $x$ doesn't seem to be a polynomial-type. Maybe "enough approximated" is the term you meant (although "enough" is a matter of discussion, indeed).
  **Response:** The wording is changed

6 Discussion

- line 326: "floatation criterion" $=>$ "hydrostatic floatation criterion"
  **Response:** The correction has been made.

- line 326: "Depending on how many of the nodes that are floating, the amount of friction in the triangle is determined." Maybe this could be rewritten. Basically, the amount of friction is computed according to the grounded are in partly floating elements. And Seroussi et al., 2014, used different techniques, based on the FEM, to compute this amount of friction.
  **Response:** The paragraph is rewritten with more details from Seroussi's paper ().

- line 327: "Also, a higher order polynomial integration over the triangles in FEM allows an inner structure in the triangular element." I didn't understand this phrase; what does it mean "inner structure" here?
  **Response:** See above.

- line 330: "... If $\chi \geq 0$ ..." $\chi$ is higher than 0 if the water pressure is computed using the bedrock elevation ($b(x)$, as described in line 269). Otherwise, $\chi = 0$ on the floating nodes (as in line 148, unless in the case as shown in Fig. 6).
  **Response:** We don't quite understand this comment. So, nothing is changed.

- line 337: "model inaccuracy" $=>$ "numerical error of the model"
  **Response:** The correction has been made.

- line 340: The subgrid also helps the convergence in comparison to non-subgrid scheme. See for example the comparison between NSEP and SEP1 or SEP2 in Seroussi et al., 2014 (e.g., Fig 2)
  **Response:** A sentence is added in the first paragraph.

- line 340: How does your subgrid implementation compare to the "DI implementation" tested in Gagliardini et al. 2016?
  **Response:** A comparison is made in the first paragraph now.

7 Conclusions

- line 341: "Subgrid models at the GL" $=>$ "Subgrid schemes to model the GL dynamics"
  **Response:** The correction has been made.

- line 341: "3D flow" actually it is also a SSA-2D flow
  **Response:** The correction has been made.

- line 342: "3D flow" same here; BISICLES is a "2 1/2 flow", but the grid is 2D (plan x-y)
  **Response:** The correction has been made.

- line 345: "subgrid model in 2D" $=>$ "subgrid scheme in 2D vertical"
  **Response:** The correction has been made.

- line 346: Note that $\tilde{\chi}$ here is the modified version of $\chi$. Please, try to condensate and simplify the definitions along the text, and try to use just one
  **Response:** After rewritten Section 4, we only use $\tilde{\chi}$ for estimating the GL position.

- line 348: "method" $=>$ "numerical scheme"
  **Response:** The correction has been made.

- line 348: delete "in 2D"
  **Response:** The correction has been made.

- line 348: "The data" $=>$ "The model setups"
  **Response:** The correction has been made.

- line 350: "with subgrid modeling" $=>$ "using the subgrid scheme"
  **Response:** The correction has been made.

- line 351: please, delete "Without further knowledge of the basal conditions and detailed models at the GL".
  **Response:** The correction has been made.

- line 352: "... approximation of the GL position." $=>$ "...approximation of the GL position, and accelerates the GL position convergence in comparison to schemes where the GL relies just on element nodes." Note that this last phrase (suggestion) only makes sense if more numerical tests are performed, helping the convergence analysis of the proposed subgrid scheme.
  **Response:** We did not make any new numerical tests.

As a last suggestion, maybe change "subgrid model" in the title to "subgrid scheme"
**Response:** The correction has been made.

**References**

[revised manuscript text omitted]

---

## Author Comment (AC2) · 10 Dec 2019

article [english]babel amsmath amssymb color

[Figure]

**Response to Anonymous Referee #2**

December 10, 2019

This paper tries to implement the subgrid scheme of grounding line (GL) movements in Elmer/Ice and test it with the 2D MISMIP benchmark. The "full" Stokes model is computational intense, especially for solving marine ice sheet problems where very fine mesh resolution is usually needed around GL to accurately capture the movement of GL. Thus, a subgrid scheme study like this paper is certainly valuable. However, the current version of this manuscript is probably not ready yet for a consideration of publication, due to the following reasons:

1. The authors use a hydrostatic (first-order) approximation to determine whether a node is floating or grounded, which looks disappointing for a "full" Stokes model. In Elmer/Ice, to solve the contact problem of GL dynamics, the normal stress (nodal force) was actually used in previous studies (e.g., Durand et al., 2009). It's surprising that this paper doesn't use that, which, from my point of view, is not a Stokes solution.
   **Response**: The GL position is determined by linear interpolation in $\chi = \sigma_{nn} + p_w$ with $\sigma_{nn}$ given by the full Stokes solution. A first order approximation of the GL position is analyzed in Sect 2.4 by perturbation theory in (Schoof, 2011) and computations in (Nowicki and Wingham 2008) show that $\chi$ is linear in $x$ for $x <$

$x_{GL}$. The GL position in other papers solving the FS equations is located in the nodes of the mesh which is a zeroth order approximation. The basis functions in our FEM model are close to linear at the lower boundary and $b(x)$ is linear between the nodes. Hence, a linear approximation for the GL is natural.

2. For testing the capability of the subgrid scheme, the authors should at least do the MISMIP3d experiments, in order to see how the GL move in y. The MISMIP benchmark is good, but I don't understand why not trying the MISMIP3d since it's been out there for a couple of years. The authors have some discussions of extending the 2d implementation to 3d, but perhaps it's better to just test it.
**Response**: Implementing and testing the GL treatment in the present structure of Elmer/ICE would require a considerable effort of many months.

3. The writings, particularly the introduction section, still needs improvement.
**Response**: The Introduction has been shortened with fewer references to other work.

**1   Technical Comments**

- Citation style throughout the whole paper needs to be corrected
  **Response**: The correction has been made.

- In the title, there should be a space right after Elmer/ICE, or just remove (v8.3)
  **Response**: The correction has been made.

- Line 13: change it to "...an indicator of ice sheet advances or retreats"
  **Response**: The correction has been made.

- Line 15: "on West Antarctica" to "in West Antarctica"
  **Response**: The correction has been made.

[Figure]

- Line 18: In theory the Stokes model is the most accurate, but in reality it might not the one that shows the best match to observations. So please make an explicit and correct statement.
  **Response**:  We have added that the FS model is the most accurate in theory. If it is the most accurate one then the analytical solution to the equations should also have the best agreement in general with observations. This is our interpretation of an accurate model. Analytical solutions are not known and they are approximated by numerical solutions. If the FS model is not the best one in comparison with data then the numerical approximation is not sufficiently accurate in our opinion. Most other equations for ice simulation are simplifications of the FS equations where terms have been removed. If the terms are small then the solution to the simplified equations is close to the FS solution but if they are large then their solution may be closer to the observed data occasionally but in general that is not the case.

- Line 25: It is the longitudinal stress gradient, not longitudinal stress, that controls the flow of ice shelf.
  **Response**: The correction has been made.

- Line 121: change "net surface accumulation/ablation" to "surface mass balance"
  **Response**: The correction has been made.

- Line 124: change "net accumulation/ablation at the lower surface" to "basal mass balance"
  **Response**: The correction has been made.

- Line 134: change "short interval" to "short distance interval"
  **Response**: The correction has been made.

- Line 270-273: For case i, how come the GL position is at the floating part? It may look reasonable numerically, but it's totally not physical.

[Figure]

**Response**: Only a part of the element is floating with floating boundary conditions. The angle between the element and the bedrock is small in practice and exaggerated in the figure.

---

## Author Comment (AC3) · 10 Dec 2019

article [english]babel amsmath amssymb color

**Response to Anonymous Referee #3**

December 10, 2019

This paper presents a subgrid interpolation across the grounding line for the Stokes equation. This interpolation is based on the stress balance at the base of the ice, $\chi = p_w + \sigma_{nn}$ with $\sigma_{nn}$ the normal deviatoric stress at the ice base, and $p_w$ the water pressure. If the last grounded element is at node $i$, and the first floating element is at $i + 1$, then the position $x_{GL}$ of the grounding line is determined as the first point where $\chi$ goes to zero. This position is then used in the evaluation of the weak form of the basal boundary condition.

Unfortunately, the incorrect citation style and at times awkward writing of the paper (in particular the introduction) distract from its contents. These need to be corrected before publication can be considered.
**Response**: We apologize for the citation style. They have been corrected.

Scientifically, the approach is a logical first step for interpolation of the basal boundary condition across the grounding line, and the same approach has been used in Seroussi et al. (2004), though not for a Full Stokes model. That said, I think that the analysis of the results could be improved significantly by

1. comparison with other interpolation schemes in 1 horizontal dimension, and

2. extension to 2 horizontal dimensions. The latter case is briefly discussed in lines 229-333, but I think an implementation would show whether this approach in indeed able to deal with complex grounding line geometries.

These kinds of comparisons are standard for the study of numerical grounding line schemes (see e.g., Seroussi et al., 2004, Feldmann et al., 2014).

For point 1 above, I am specifically interested in seeing a comparison to an interpolation of the basal shear stress constant $\beta$ between the last grounded and the first floating point, i.e., if one would multiply $\beta$ with $(x_{GL} - x_i)/(x_{i+1} - x_i)$, how would the results differ? This is the interpolation scheme traditionally used in depth-integrated models (e.g., Feldmann et al., 2014, Pattyn et al., 2006) and also introduced as SEP1 in Seroussi et al. (2014). I am wondering whether such an interpolation alone would already provide the observed improvement in the numerical performance, as suggested from my reading of Seroussi et al. (2014). I am also sceptical about the effect of setting $\beta$ to $\beta/2$ in the interpolated cell, as suggested in lines 282-283? This introduces an additional interpolation which is similar to the interpolations used in models with a structured grid, and has no physical basis in the presented scheme as the interpolation is already done by splitting up the integral for the boundary condition.

**Response**: Actually, the subgrid model we developed is equivalent to multiplying $\beta$ by $(x_{GL} - x_i)/(x_{i+1} - x_i)$. This explanation is added in the revised version according to the comments from referee #1. However, the difficulty in this formulation is not only about how to treat $\beta$, but how to determine $x_{GL}$ in an accurate way in the FS equations. Also, as the boundary conditions in FS involve $\mathbf{u} \cdot \mathbf{n} = 0$, we introduced the Nitsche's method to weakly impose this, such that it can also be imposed partially on the GL element. The details of the implementation are rephrased in Section 4 with Fig. 2 and 3, and Algorithm 3.

Moreover, it wasn't completely clear to me which parts of the FEM implementation in Elmer/Ice were actually altered. For example, my impression of the time-stepping

scheme is that it is basically the same as in Durand et al. (2009a), in which case section 3.3. is unnecessary.

**Response**: Section 3.3 is included to: 1. explain the whole algorithm in which the GL treatment is embedded, 2. interpret the damping introduced in (Durand 2009a) as an implicit time integration of the position of the floating ice base, 3. show in a simple calculation how short the timesteps would be with an explicit method.

Other comments (kept short as I think the entire paper needs to be rewritten):

1. Better use $\dot\epsilon$ for the strain rate, $\tau$ is usually used for a stress tensor
   **Response**: The correction has been made.

2. Equations (7) and (8) are the kinematic boundary conditions, and should be referred to as such
   **Response**: We agree that they are the kinematic boundary conditions. However, these names follow the convention as in (; ).

**References**

Gagliardini, O., Zwinger, T., Gillet-Chaulet, F., Durand, G., Favier, L., de Fleurian, B., Greve, R., Malinen, M., Martín, C., Råback, P., Ruokolainen, J., Sacchettini, M., Schäfer, M., Seddik, H., and Thies, J.: Capabilities and performance of Elmer/Ice, a new generation ice-sheet model, Geosci. Model Dev., 6, 1299–1318, 2013.

Pattyn, F., Schoof, C., Perichon, L., Hindmarsh, R. C. A., Bueler, E., de Fleurian, B., Durand, G., Gagliardini, O., Gladstone, R., Goldberg, D., Gudmundsson, G. H., Huybrechts, P., Lee, V., Nick, F. M., Payne, A. J., Pollard, D., Rybak, O., Saito, F., and Vieli, A.: Results of the Marine Ice Sheet Model Intercomparison Project, MISMIP, Cryosphere, 6, 573–588, 2012.

---

## Author Comment (AC4) · 10 Dec 2019

We uploaded the marked changes in the supplement here: https://editor.copernicus.org/index.php?_mdl=msover_md&_jrl=365&_lcm=oc108lcm109w&_acm=get_comm_sup_file&_r

---

## Author Response (AR1)

Dear Dr. Alex,

We have revised the manuscript according to the comments from the three referees. As you may notice, referee #1 generously provided 9 pages comments, and the other two referees' comments are more or less covered by it. Then, we did not bother to mark the changes separately for the three referees.

We uploaded the marked changes in the supplement of the reply to referee 1, here: here

And the revised manuscript is uploaded in ''File upload'.

We really appreciate your help.

Best wishes,

[revised manuscript text omitted]

---

## Referee Report (RR1)

**Review Cheng, et al., A full Stokes subgrid scheme for simulation of grounding line migration in ice sheets using Elmer/ICE (v8.3)**

**Review based on the file named: *gmd-2019-244-manuscript-version4.pdf**

The manuscript aims to present a numerical scheme to deal with the friction inside elements partly floating in a (full-)Stokes formulation for the marine ice sheet simulation. The formulation and results are carried out in a 2D vertical domain, and possible extension to 3D domain is discussed. The reviewed version presents the corrections asked in the first review, mainly in the technical part (methods). The Introduction was changed, but additional "polishing" is needed before publishing. No additional simulations were carried out, and the presentation of the results was not modified.

**General comments:**

- The numerical scheme is better presented in this new version, although some minor corrections should be done. See specific comments.
- With the results presented along the manuscript it is hard to analyze the convergence (and consistency) of the subgrid scheme proposed. There is no convergence rate analysis or comparison with the cited reference work (Gagliardini et al., 2016). I strongly recommend additional simulations (mesh resolutions equal to 500 m and 250 m) and a comparison with the results from Gagliardini et al. (2016), mainly in terms of GL position against mesh resolution.
- The overall explanation of the subgrid scheme was improved, which helps the reproducibility of the results.
- The Introduction section must be improved yet. The reading is not smooth yet, and additional polishing is needed to make the reading "pleasant" enough for a scientific/technical paper.
- The citation style along the manuscript was corrected, but there are still corrections in some parts. See specific comments.

**Specific comments:**

- line 77: "for modeling of the flow" => "for modeling the flow"

- line 78: "These nonlinear" => "The nonlinear"

- line 102: "$\beta$" => "$\beta$ ($\geq 0$)"

- line 117: "$z_b < 0$" => "$z_b > b(x)$"

- line 119: "The solution close to the grounding line" => "A first order solution close to the grounding line" or "A solution close to the grounding line from the boundary layer theory" or "A boundary layer' solution close to the grounding line". Note that this solution is based on a linear Stokes problem (i.e., $n = 1$ in Glen's flow law).

- line 121: "(Schoof, 2011)" => "Schoof (2011)"

- line 123: "$u$" => "the ice velocity $u$"

- line 124: "ice surface slope is continuous": are you referring to slope or just the ice surface? Does this proposition come from Schoof (2011)? Also, why this is important/relevant for the subgrid scheme used here?

- line 128: "(Durand et al., 2009a)" => "Durand et al. (2009a)"

- line 129: "(Schoof, 2011, Ch. 4.3)" => "Schoof (2011, Sect. 4.3)"

- line 129: "parameters" => "parameters,"

- line 133: "variables satisfy" => "variables satisfy (Schoof, 2011)" (if the citation is right)

- line 142: "(Norwicki and Wingham, 2008)" => "Norwicki and Wingham (2008)"

- line 143: "original variables": what does it mean?

- line 149: The definition of "$k$" and "$k^{**}$" is weird. Why does the approximation space depend on the Glen's flow law? Are these not referred to the polynomial order of the space? Please, check the definition and notation of these spaces.

- line 152: Please, change the citation style here

- line 156: the form "$b(v, p)$" is not defined here (although it follows $b(u, q)$ )

- line 156: where is $\sigma_{nt}$ in the expressions? Please, check the forms $B_\Gamma$ and $B_N$

- line 156: How the forcing term ($F(v)$) is numerically considered in the element crossed by the grounding line? There is no mention of this along the text.

- line 171: Do you also split the in integral of the forcing term ($F(v)$))?

- line 173: Eq. (15): the forms $B_\Gamma$ and $B_N$ are already integrated. Please, fix the notation here.

- line 173: Eq. (15): where is the $\sigma_{nt}$? Please, check the forms here.

- line 173: Eq. (15): the forcing ($p_w n \cdot v$) is considered here, but is it included in the stiff matrix? Please, could you make it clearer?

- lines 175-177: "With a strong formulation … into account". This is phrase is not clear. I don't understand why strong formulation is mentioned here.

- line 177: "no basis functions satisfies …". I am not sure if this is true. There are lots of FEM schemes where the discontinuity is well accommodated (e.g., xFEM, CutFEM, etc). The phrase is only true if the standard FEM is used, and no specific refinement is made in the element crossed by the grounding line (as is the case of this paper).

- lines 185-186: what does "along the slope" mean?

- line 195: "The nonlinear equations …" => "The nonlinear equations, Eq. (14), …"

- line 197: "timestep" => "time step" (and elsewhere)

- line 198: "nonlinear iterations" => "nonlinear iterations (Picard)"

- Algorithm 1: All grounded nodes are marked as "GL nodes"? Please, could you make it clearer along the text? Also, check the text punctuation in Algorithm 1

- Algorithm 2: please, check the text punctuation in Algorithm 2

- line 208: "A stability problem" = > "A numerical stability problem"

- line 208: "(Durand et al., 2009a)" => "Durand et al. (2009a)" Same in lines 209, 238.

- line 215: "is updated implicitly": is $p_w$ also considered in the forcing term of Eq. (14)? Could you make it clearer?

- line 221: note that "$n$" was used before with another meaning

- line 242: "(Seroussi et al., 2014)" => "Seroussi et al., (2014)". The same for Schoof citation

- line 249: "(11)" => "Eq. (11)"

- line 251: please, change the citation style here

- line 251: "analytical solution": which one? From Schoof 2011's paper? Same in line 254. Note that if it is from Schoof (2011), it is based on a linear Stokes problem $n = 1$ (Glen's flow law).

- line 258: "between" => "between any"

- line 261: "basal surface of the ice" => "ice base"

- line 266: "external forces" => "external forces and boundary conditions" (maybe?)

- line 267: "geometrically grounded": how is the element identified as geometrically grounded or geometrically floating, in the numerical framework? Could you make it clearer along the text?

- line 269: "(Gagliardini et al., 2016)" => "Gagliardini et al. (2016)"

- line 270: please, delete the extra "the"

- line 271: "fine mesh" => "fine mesh resolution (<100 m)" (maybe?)

- Fig. 2 and Fig. 3: "net forces" => "net forces in the vertical direction" (please, check also the text)

- Eq. (27): $\chi(x_i) = 0$, right? Or this is not zero in the numerical solution? Please, could you make it clearer along the text?

- line 280: "best numerical approximation". I don't know if "best" is the word here. Maybe mentioning that it is in the same order of the framework/scheme/approximation space

- line 284-285: "Considering … always stays": maybe this phrase is unnecessary; even the numerical GL position stays on bedrock

- line 293: "bottom surface" => "ice base"

- line 296: please, delete "Then"

- line 297: "(Seroussi et al., 2014)" => "Seroussi et al., (2014)". Same in line 299

- line 299: "condition" => "condition, respectively"

- line 299: "reasonable resolution" => "reasonable numerical accuracy"

- line 300: "required" => "used". The integration points are defined over the GL element, right? And the step function makes the work of selecting the area to be integrated, right? Then, note that, depending on the situation, even a tenth order could not be enough to carry out the integration with enough numerical accuracy (as is the SEP2 method of Seroussi et al., 2014, where the distribution of the integration points follows the grounding line position inside the GL element). Besides that, the approach used here seems reasonable, and it is easier to be implemented in comparison to SEP2-type scheme.

- line 302: "fully on the ground": geometrically, right?

- line 304: "basal surface" => "ice base"

- line 307: "fully grounded" => "fully geometrically grounded" (maybe?)

- line 308: "boundary elements" => "basal elements" (maybe?)

- line 311: "floating elements" => "fully geometrically floating elements" (maybe)

- line 313: "grounded" => "geometrically grounded" (maybe)

- line 313: "analytical solution": maybe "numerical solution"? It is not clear what you meant here

- line 315: "3" => "Fig. 3"

- Eq. (29): check the notation of the forms $B_\Gamma$ and $B_N$. Also, there is no $\sigma_{tn}$ here

- lines 319-320: How are the phases (advance or retreat) defined? Comparing with previous (last time step) GL position? Please, could you make it clearer?

- Algorithm 3: please, check the text punctuation in Algorithm 3

- line 326: in the calculation of $\tilde{\chi}$, $p_w$ is kept constant, right? Could you please make it clearer?

- line 330: "(Gagliardini et al., 2016)" => Gagliardini et al., (2016)". The same in lines 331, 367, 369, 370, 372, 397, 398

- line 342: "both for" => "for both"

- line 344: please, correct the citation style here

- line 352: "(van Dongen et al., 2018)" => "van Dongen et al. (2018)"

- line 357: "(Schoof, 2011)" => "Schoof (2011)"

- line 357: "represented" => "captured"

- line 359: "Seroussi et al (Seroussi et al., 2014)" => Seroussi et al. (2014) (maybe?)

- Fig. 6: Note that the GL is close to a node. I suspect the same is observed for the other resolutions (2 and 4 km). So, the GL position also depends on the distribution of the nodes in 1D.

- line 363: "floatation criterion" => "hydrostatic floatation criterion"

- line 372: "asymptote" => "convergence asymptote" (maybe?)

- lines 374-375: "but the numerical solution of the velocity field, pressure as well as the two free surfaces are still determined by the coarse mesh …": note that small bedrock features impact the GL dynamics, and they are important in short time scale simulations (decades). In general, mesh resolution equal to 500 m is required to capture these bedrock features near the GL. Also, from figures 6 and 7, there are expressive changes in the fields near the GL (thickness, surface, horizontal and vertical velocities). These changes are only "well" captured with enough mesh resolution (<1 km or less). Besides that, no error estimator was used here; therefore, the term "determined" doesn't fit here. The subgrid scheme tends to accelerate the rate of convergence in comparison to NSEP-type schemes (by decreasing the numerical error of one source, the boundary condition at the base), but relatively fine mesh resolution (I would say 500 m) is yet required around the GL to numerical error control (from other sources, e.g., bedrock geometry, intrinsic solutions variations around GL, effect of ocean-induced basal melting, etc).

- line 377: "following way" => "following way (considering linear Lagrange functions)" (maybe?)

- line 382: "An alternative to a subgrid scheme is to introduce dynamic adaptation of the mesh": I don't think mesh adaptation is an alternative, strictly speaking. They are complementary to each other. The subgrid scheme tends to decrease the error on the boundary condition, accelerating the rate of convergence (ideally); the mesh adaptation helps save computation effort, since enough mesh resolution (~500 m) is needed around the GL. They can (should) be used together, indeed.

- line 383: please, correct the citation style here

- line 386: "shorter timesteps are necessary for stability when the mesh size is smaller in a mesh adaptive method" => "shorter time steps are necessary for numerical stability in dynamic mesh adaptation schemes". Note that it depends on the numerical implementation; some schemes are more stable than others.

- line 387: "Introducing a time dependent mesh adaptivity into an existing code requires a substantial coding effort and will increase the computational work considerably. Subgrid modeling is easier to implement and the increase in computing time is small." I don't totally agree here. Yes, mesh adaptivity is a substantia coding effort, and there are drawbacks in scalability. But at the end, the computational effort is (or should be) much less in comparison to a fine uniform mesh. The improvement of a subgrid scheme for the basal condition (friction) makes the 25 m-mesh resolution requirement to a 500 m-mesh resolution requirement. But yet, a 500 m-mesh resolution is expressively fine in comparison to a typical horizontal scale of ice sheets (order of 1,000 km). A static mesh adaptation (performed during the domain discretization) could be used instead of dynamic mesh adaptation (considering the GL will not migrate beyond the adapted/refined region). For short-term simulations (decades) this is feasible, but this is not totally true for paleo-ice sheets simulations. Therefore, using subgrid scheme with dynamic mesh adaptation should work properly (in the sense of convergence of the GL dynamics with reduced computational effort).

- lines 390-392: "A subgrid scheme … (Feldmann et al., 2014)": this phrase could be migrated to the discussion part. Also, correct the citation style here.

- line 395: "function $\chi(x)$" => "function $\chi(x)$ based on a first order approximation of the basal stress balance" (maybe?) Again, note that the solution from Schoof (2011) considers $n = 1$ (Glen's flow law), as you have well pointed along the text.

- line 396: "is modified" => "is modified to accommodate the discontinuities in the boundary conditions"

- line 399: "Solving for … GL position": I think this phrase could be deleted.

---

## Editor Decision (ED1)

[revised manuscript text omitted]

*Handwritten annotations (red):* There is no convergence study as you indicate in the reviewer response, but they doesn't appear to be monotonic approach to a solution.

[revised manuscript text omitted]

---

## Author Response (AR2)

Dear Dr. Alex,

We have responded to all the comments by the referees. The advance and retreat solutions have been computed with a 0.5 km long spatial step as suggested by a referee. The grounding line (GL) position does not move very much even compared to the 4 km step but the velocity is much better resolved at the GL with the finer mesh. We write in the title and Introduction that the scheme is tested in two dimensions but it can be extended to three dimensions as sketched in Discussion. The GL is found by a first order accurate numerical method which is consistent with all other first order approximations in the finite element method used in this paper and Elmer/ICE. We review other subgrid methods for other models which all are simplifications of the full Stokes (FS) model. The FS model is needed at the GL to capture the vertical stress component there. We refer to papers with arguments supporting that view. The difficulty with the FS equations compared to e.g. the SSA equations is that the vertical velocity in FS moves the base of the floating ice and Archimedes' flotation criterion is not valid at the GL. The vertical velocity introduces another boundary condition on the velocity in the normal direction of the grounded ice which disappears after the GL. The main result is that by subgrid modeling we obtain an accuracy for the GL position comparable to previously published results using more than 20 times larger spatial steps (25-200 m).

We really appreciate your help.

Best wishes,

Gong Cheng, Per Lötstedt and Lina von Sydow

**Response to Anonymous Referee #1**

**March 10, 2020**

The manuscript aims to present a numerical scheme to deal with the friction inside elements partly floating in a (full-)Stokes formulation for the marine ice sheet simulation. The formulation and results are carried out in a 2D vertical domain, and possible extension to 3D domain is discussed. The reviewed version presents the corrections asked in the first review, mainly in the technical part (methods). The Introduction was changed, but additional "polishing" is needed before publishing. No additional simulations were carried out, and the presentation of the results was not modified

**General comments**

• The numerical scheme is better presented in this new version, although some minor corrections should be done. See specific comments.

**Response**: The correction has been made.

• With the results presented along the manuscript it is hard to analyze the convergence (and consistency) of the subgrid scheme proposed. There is no convergence rate analysis or comparison with the cited reference work (Gagliardini et al., 2016). I strongly recommend additional simulations (mesh resolutions equal to 500 m and 250 m) and a comparison with the results from Gagliardini et al. (2016), mainly in terms of GL position against mesh resolution.

**Response**: Additional simulation with 500 m mesh resolution has been added, and it is compared with the results in Gagliardini et al. (2016), shown in a new Fig. 6.

• The overall explanation of the subgrid scheme was improved, which helps the reproducibility of the results.

**Response**: Thanks for the comments. We improved the subgrid section a bit more according to the specific comments.

• The Introduction section must be improved yet. The reading is not smooth yet, and additional polishing is needed to make the reading "pleasant" enough for a scientific/technical paper.

**Response**: The Introduction has been modified.

• The citation style along the manuscript was corrected, but there are still corrections in some parts. See specific comments.

**Response**: The corrections have been made.

**Specific comments**

- line 77: "for modeling of the flow" => "for modeling the flow"
   **Response**: The correction has been made.
- line 78: "These nonlinear" => "The nonlinear"
   Response: The correction has been made.
- line 102: "β" => "β(≥ 0)"
   Response: The correction has been made.
- line 117: zb < 0 => zb > b(x) **Response**: The correction has been made.
- line 119: "The solution close to the grounding line" => "A first order solution close to the grounding line" or "A solution close to the grounding line from the boundary layer theory" or "A boundary layer' solution close to the grounding line". Note that this solution is based on a linear Stokes problem (i.e., n = 1 in Glen's flow law).

**Response**: The headline has been changed.

• line 121: "(Schoof, 2011)" => "Schoof (2011)"

**Response**: The correction has been made.

• line 123: "u" => "the ice velocity u"

**Response**: The correction has been made.

• line 124: "ice surface slope is continuous": are you referring to slope or just the ice surface? Does this proposition come from Schoof (2011)? Also, why this is important/relevant for the subgrid scheme used here?

**Response**: These are the words used by Schoof. We could have written 'the space derivative of the height of the ice is continuous' but chose to do it in this way. A reference to Schoof's paper is included. The section describes the analytically derived properties of the solution close to the GL. We show later how our treatment of the GL agrees to first order with the analysis by Schoof.

line 128: "(Durand et al., 2009a)" => "Durand et al. (2009a)"
Response: The correction has been made.

- line 129: "(Schoof, 2011, Ch. 4.3)" => "Schoof (2011, Sect. 4.3)"
  Response: The correction has been made.
- line 129: "parameters" => "parameters,"
   Response: The correction has been made.
- line 133: "variables satisfy" => "variables satisfy (Schoof, 2011)" (if the citation is right)

**Response**: The correction has been made.

 line 142: "(Norwicki and Wingham, 2008)" => "Norwicki and Wingham (2008)"

**Response**: The correction has been made.

• line 143: "original variables": what does it mean?

**Response**: The original, unscaled variables is what it is meant. The words are removed.

• line 149: The definition of "k" and "k\*" is weird. Why does the approximation space depend on the Glen's flow law? Are these not referred to the polynomial order of the space? Please, check the definition and notation of these spaces.

**Response**: We now tell where in the three theoretical papers the functional spaces are defined. They are the same in all the three papers. A short discussion is found in the paper by Jouvet and Rappaz(2011). We do not delve into this issue in details in our paper.

• line 152: Please, change the citation style here

**Response**: The correction has been made.

• line 156: the form "b(v,q)" is not defined here (although it follows b(u,p))

**Response:** As  $b(\cdot, \cdot)$  defines a bi-linear form, we think that b(v, p) follows from the definition of b(u, q) in Eq (14).

• line 156: where is  $\sigma_{nt}$  in the expressions? Please, check the forms  $B_{\Gamma}$ ; and  $B_N$

**Response**:  $\sigma_{nt}$  is now in the definition of  $B_{\Gamma}$ . On  $\Gamma_{bg}$ ,  $\sigma_{nt}$  is replaced by the slip boundary condition  $-\beta u$ . On  $\Gamma_{bf}$ ,  $\sigma_{nt} = 0$ .

• line 156: How the forcing term F(v) is numerically considered in the element crossed by the grounding line? There is no mention of this along the text.

**Response:** The forcing term is split into the interior term F and the boundary term  $F_{\Gamma}$ . A sentence is added to explain the boundary term  $F_{\Gamma}(v)$ . The integration of  $F_{\Gamma}$  is shown Eqs (15) and (30) in the numerical Section 4.

• line 171: Do you also split the in integral of the forcing term F(v)?

**Response:** As explained above, the term F(v) is from the gravitational force which is smooth over the GL. The boundary term  $F_{\Gamma}$  is from the ice-ocean interface and is integrated partially on the floating ice as in Eqs (15) and (30).

• line 173: Eq. (15): the forms  $B_{\Gamma}$ ; and  $B_N$  are already integrated. Please, fix the notation here.

**Response**: This is corrected now.

- line 173: Eq. (15): where is the  $\sigma_{nt}$ ? Please, check the forms here.
- **Response:** As explained above (line 156),  $\sigma_{nt}$  is replaced by the slip boundary condition in  $B_{\Gamma}$  on  $\Gamma_{bg}$  and vanishes due to the ice-ocean interface condition on  $\Gamma_{bf}$ .
- line 173: Eq. (15): the forcing  $p_w n \cdot v$  is considered here, but is it included in the stiff matrix? Please, could you make it clearer?

**Response:** No,  $p_w n \cdot v$  is not included in the stiffness matrix because it does not depend on u.

• lines 175-177: "With a strong formulation ... into account". This is phrase is not clear. I don't understand why strong formulation is mentioned here.

**Response**: We mean the strong formulation of the boundary condition. We have added more explanation after Eq (15).

• line 177: "no basis functions satisfies ...". I am not sure if this is true. There are lots of FEM schemes where the discontinuity is well accommodated (e.g., xFEM, CutFEM, etc). The phrase is only true if the standard FEM is used, and no specific refinement is made in the element crossed by the grounding line (as is the case of this paper).

**Response**: Yes, this is correct. We remark that we use the standard FEM basis functions in Elmer/ICE after Eq (28).

• lines 185-186: what does "along the slope" mean?

**Response**: It is changed to "along the ice base".

• line 195: "The nonlinear equations ..." => "The nonlinear equations, Eq. (14), ..."

- line 197: "timestep" => "time step" (and elsewhere)
   Response: The correction has been made.
- line 198: "nonlinear iterations" => "nonlinear iterations (Picard)"
   Response: The correction has been made.

• Algorithm 1: All grounded nodes are marked as "GL nodes"? Please, could you make it clearer along the text? Also, check the text punctuation in Algorithm 1

**Response**: The term "GL nodes" is only used in Algorithm 1. A new sentence is added before Algorithm 1 explaining that it will be in one element. Some text punctuation is added.

- Algorithm 2: please, check the text punctuation in Algorithm 2 **Response**: The correction has been made.
- line 208: "A stability problem" => "A numerical stability problem"
   Response: The correction has been made.
- line 208: "(Durand et al., 2009a)" => "Durand et al. (2009a)" Same in lines 209, 238.

**Response**: The correction has been made.

• line 215: "is updated implicitly": is  $p_w$  also considered in the forcing term of Eq. (14)? Could you make it clearer?

**Response**: Yes, the  $p_w$  in Eq (14) is udated implicitly. A few words after Eq (23) explain this.

• line 221: note that n was used before with another meaning

**Response**: All the terms with n for the time discretization are changed to  $\ell$ .

• line 242: "(Seroussi et al., 2014)" => "Seroussi et al., (2014)". The same for School citation

**Response**: The correction has been made.

- line 249: "(11)" => "Eq. (11)"
   Response: The correction has been made.
- line 251: please, change the citation style here **Response**: The correction has been made.
- line 251: "analytical solution": which one? From Schoof 2011's paper? Same in line 254. Note that if it is from Schoof (2011), it is based on a linear Stokes problem n = 1 (Glen's flow law).

**Response**: It is the analytical solution to the FS equation, without any approximation or simplification. This is now more clearly stated.

• line 258: "between" => "between any"

• line 261: "basal surface of the ice" => "ice base"

**Response**: The correction has been made.

• line 266: "external forces" => "external forces and boundary conditions" (maybe?)

**Response**: The correction has been made.

• line 267: "geometrically grounded": how is the element identified as geometrically grounded or geometrically floating, in the numerical framework? Could you make it clearer along the text?

**Response**: A reference to Algorithm 1 is added to clarify this.

- line 269: "(Gagliardini et al., 2016)" => "Gagliardini et al. (2016)" Response: The correction has been made.
- line 270: please, delete the extra "the"
   Response: The correction has been made.
- line 271: "fine mesh" => "fine mesh resolution (< 100 m)" (maybe?)</li>
   Response: The correction has been made.
- Fig. 2 and Fig. 3: "net forces" => "net forces in the vertical direction" (please, check also the text)

**Response**: The correction has been made.

• Eq. (27):  $\chi(x_i) = 0$ , right? Or this is not zero in the numerical solution? Please, could you make it clearer along the text?

**Response:** The extrapolated  $\chi$  and  $\tilde{\chi}$  satisfy  $\chi(x_i) > 0$  and are better explained now in Sect 4. A modification of them is necessary at  $x_i$ .

• line 280: "best numerical approximation". I don't know if "best" is the word here. Maybe mentioning that it is in the same order of the framework/scheme/approximation space

**Response**: We have written about it below Eq (29).

• line 284-285: "Considering ... always stays": maybe this phrase is unnecessary; even the numerical GL position stays on bedrock

**Response**: We have removed the sentence. It follows from the interpolation in Eq (28) that the numerical  $x_{GL}$  stays on the element boundary.

• line 293: "bottom surface" => "ice base"

**Response**: The correction has been made.

• line 296: please, delete "Then"

 line 297: "(Seroussi et al., 2014)" => "Seroussi et al., (2014)". Same in line 299

**Response**: The correction has been made.

- line 299: "condition" => "condition, respectively"
   Response: The correction has been made.
- line 299: "reasonable resolution" => "reasonable numerical accuracy"
   **Response**: The correction has been made.
- line 300: "required" => "used". The integration points are defined over the GL element, right? And the step function makes the work of selecting the area to be integrated, right? Then, note that, depending on the situation, even a tenth order could not be enough to carry out the integration with enough numerical accuracy (as is the SEP2 method of Seroussi et al., 2014, where the distribution of the integration points follows the grounding line position inside the GL element). Besides that, the approach used here seems reasonable, and it is easier to be implemented in comparison to SEP2-type scheme.

**Response**: Changed. Yes, with this integration scheme, a tenth order polynomial is used to approximate the step function.

- line 302: "fully on the ground": geometrically, right?
   Response: Yes, it is changed to "fully geometrically on the ground".
- line 304: "basal surface" => "ice base"

**Response**: The correction has been made.

- line 307: "fully grounded" => "fully geometrically grounded" (maybe?)
   Response: The correction has been made.
- line 308: "boundary elements" => "basal elements" (maybe?)
   Response: The correction has been made.
- line 311: "floating elements" => "fully geometrically floating elements" (maybe)

**Response**: The correction has been made.

- line 313: "grounded" => "geometrically grounded" (maybe)
   Response: The correction has been made.
- line 313: "analytical solution": maybe "numerical solution"? It is not clear what you meant here

**Response**: The analytical solution to the FS.

• line 315: "3" => "Fig. 3"

**Response**: The correction has been made.

• Eq. (29): check the notation of the forms  $B_{\Gamma}$  and  $B_N$ . Also, there is no  $\sigma_{nt}$  here

**Response:** The notation has been changed and  $\sigma_{nt}$  is replaced by  $-\beta(t \cdot u)(t \cdot v)$  as in Eq (14).

• lines 319-320: How are the phases (advance or retreat) defined? Comparing with previous (last time step) GL position? Please, could you make it clearer?

**Response**: This is clarified now in Sect 4.

- Algorithm 3: please, check the text punctuation in Algorithm 3 **Response**: The correction has been made.
- line 326: in the calculation of  $\tilde{\chi}, p_w$  is kept constant, right? Could you please make it clearer?

**Response**: Yes, it is fixed. A new sentence is added.

- line 330: "(Gagliardini et al., 2016)" => Gagliardini et al., (2016)". The same in lines 331, 367, 369, 370, 372, 397, 398
   **Response**: The correction has been made.
- line 342: "both for" => "for both"
   **Response**: The correction has been made.
- line 344: please, correct the citation style **Response**: The correction has been made.
- line 352: "(van Dongen et al., 2018)" => "van Dongen et al. (2018)"
  Response: The correction has been made.
- line 357: "(Schoof, 2011)" => "Schoof (2011)"
   Response: The correction has been made.
- line 357: "represented" => "captured
   Response: The correction has been made.
- line 359: "Seroussi et al (Seroussi et al., 2014)" => Seroussi et al. (2014) (maybe?)

• Fig. 6: Note that the GL is close to a node. I suspect the same is observed for the other resolutions (2 and 4 km). So, the GL position also depends on the distribution of the nodes in 1D.

**Response**: Yes, it depends on the node positions and the mesh size but that is true also for the smooth solution away from the GL. The solution is mesh dependent. The GL position between the nodes is mentioned in the end of Results.

- line 363: "floatation criterion" => "hydrostatic floatation criterion" Response: The correction has been made.
- line 372: "asymptote" => "convergence asymptote" (maybe?)

**Response**: The correction has been made.

• lines 374-375: "but the numerical solution of the velocity field, pressure as well as the two free surfaces are still determined by the coarse mesh ...": note that small bedrock features impact the GL dynamics, and they are important in short time scale simulations (decades). In general, mesh resolution equal to 500 m is required to capture these bedrock features near the GL. Also, from figures 6 and 7, there are expressive changes in the fields near the GL (thickness, surface, horizontal and vertical velocities). These changes are only "well" captured with enough mesh resolution (1) km or less). Besides that, no error estimator was used here; therefore, the term "determined" doesn't fit here. The subgrid scheme tends to accelerate the rate of convergence in comparison to NSEP-type schemes (by decreasing the numerical error of one source, the boundary condition at the base), but relatively fine mesh resolution (I would say 500 m) is yet required around the GL to numerical error control (from other sources, e.g., bedrock geometry, intrinsic solutions variations around GL, effect of ocean-induced basal melting, etc).

**Response**: We have new Figs. 7 and 8 with 500 m resolution. Except for the GL position, the solution around the GL looks very much the same (excluding details) as the solution with 1 km resolution. Larour et al(2019) say that 1 km is satisfactory. It seems as if 500 m is sufficient. We have added one new reference where the sensitivity to the base friction and the bedrock geometry is investigated. The sensitivity increases the closer the surface observation of velocity and height is to the GL.

• line 377: "following way" => "following way (considering linear Lagrange functions)" (maybe?)

**Response**: The correction has been made.

• line 382: "An alternative to a subgrid scheme is to introduce dynamic adaptation of the mesh": I don't think mesh adaptation is an alternative, strictly speaking. They are complementary to each other. The subgrid scheme tends to decrease the error on the boundary condition, accelerating

the rate of convergence (ideally); the mesh adaptation helps save computation effort, since enough mesh resolution (500 m) is needed around the GL. They can (should) be used together, indeed.

**Response**: We have rewritten the paragraph now discussing static and dynamic adaptation and subgrid modeling.

• line 383: please, correct the citation style here

**Response**: The correction has been made.

line 386: "shorter timesteps are necessary for stability when the mesh size is smaller in a mesh adaptive method" => "shorter time steps are necessary for numerical stability in dynamic mesh adaptation schemes". Note that it depends on the numerical implementation; some schemes are more stable than others.

**Response**: The correction has been made.

• line 387: "Introducing a time dependent mesh adaptivity into an existing code requires a substantial coding effort and will increase the computational work considerably. Subgrid modeling is easier to implement and the increase in computing time is small." I don't totally agree here. Yes, mesh adaptivity is a substantial coding effort, and there are drawbacks in scalability. But at the end, the computational effort is (or should be) much less in comparison to a fine uniform mesh. The improvement of a subgrid scheme for the basal condition (friction) makes the 25 m-mesh resolution requirement to a 500 m-mesh resolution requirement. But yet, a 500 m-mesh resolution is expressively fine in comparison to a typical horizontal scale of ice sheets (order of 1,000 km). A static mesh adaptation (performed during the domain discretization) could be used instead of dynamic mesh adaptation (considering the GL will not migrate beyond the adapted/refined region). For short-term simulations (decades) this is feasible, but this is not totally true for paleo-ice sheets simulations. Therefore, using subgrid scheme with dynamic mesh adaptation should work properly (in the sense of convergence of the GL dynamics with reduced computational effort).

**Response**: The paragraph has been rewritten as described above taking these comments into account.

• lines 390-392: "A subgrid scheme ... (Feldmann et al., 2014)": this phrase could be migrated to the discussion part. Also, correct the citation style here.

**Response**: We have decided to keep the work by other researchers on subgrid schemes in Conclusions to contrast them to our work.

• line 395: "function  $\chi(x)$ " => "function  $\chi(x)$  based on a first order approximation of the basal stress balance" (maybe?) Again, note that the

solution from Schoof (2011) considers n = 1 (Glen's flow law), as you have well pointed along the text.

**Response:** The functions  $\chi(x)$  and  $\tilde{\chi}(x)$  are nonlinear. With the FEM discretization and linear Lagrange element we use, they are piecewise linear in x. This is remarked in Sect 4 now. In an expansion in small parameters and taking the first order approximation for n = 1 by Schoof we obtain  $\chi_a$  which is close to our linear approximation.

• line 396: "is modified" => "is modified to accommodate the discontinuities in the boundary conditions"

**Response**: The correction has been made.

• line 399: "Solving for ... GL position": I think this phrase could be deleted.

**Response to Anonymous Referee #2**

March 11, 2020

**Major concerns**

1. As clearly illustrated in the paper (Section 2.4, and many other places), the authors are using a first-order approximation to determine the location of the grounding line. Thus, I don't quite understand why they call it a "full Stokes subgrid scheme"?

**Response:** The  $\chi$  and chi functions in Eqs (27) and (29) that we use to determine the GL position are nonlinear. After FEM discretization with the linear Lagrange element, they vary linearly in x over the GL element. These are the  $\chi$ -functions that we have access to. We write about this in a revised Sect. 4. The linear interpolation to find  $x_{GL}$  is consistent with the level of approximation by FEM. The discussion in Sect. 2.4 is there only to lend analytical support and inspiration to our choice of  $\chi$ . We still think that our method is a subgrid scheme for the FS equations.

2. In Elmer/Ice, originally, the location of the grounding line is decided by comparing the water pressure  $(p_w)$  and the normal stress  $(tau; which can be determined from the Stokes solution) at each node <math>(N = p_w - \tau)$ . We can tell which node is floating or grounded by looking at the sign of N. Note that Elmer/Ice uses nodal force and contact force, instead of the actual water pressure and normal stress. Therefore, if we consider a 2D case, I guess it would still be possible to estimate the exact grounding line location by simply interpolating the N values at two neighboring nodes. Did the authors test it and then decide to go for their first-order method? If yes, what is the difference between these two methods?

**Response**: For linear Lagrange elements used in this paper and Elmer/ICE, the pressure and stress can be represented by the nodal force and contact force together with the basis functions. If  $N = \tau_{nn} - p + p_w$ , this paper follows the same criteria as Elmer/ICE to determine the grounded and floating nodes. Indeed, the GL position is determined by the linear interpolation of the N value. However, notice that a naive linear interpolation will not give a good estimate of the GL position, simply because N = 0 on any floating nodes. That is why we introduce the function  $\tilde{\chi}$  and use it to determine the GL instead.

3. If the author cannot provide the sub-grid results of 3D experiments (i.e., MISMIP3d), I would question the applicability of this 2D scheme in 3D cases. I agree the 2D results are still valuable in some senses, but for a complete and thorough evaluation of this first-order sub-grid scheme, I would suggest the authors do the MISMIP3d tests before its final publication. However, if the editor and other reviewers feel the 2D experiments are sufficient to prove its applicability, I would strongly suggest they remove all 3D discussions in the paper, and explicitly demonstrate that it is a 2D scheme in the title and elsewhere in the manuscript. A possible title would be "A two-dimensional first-order subgrid scheme for simulation of grounding line migration in ice sheets using Elmer/ICE (v8.3)". By that it is safe to limit this paper in 2D discussion.

**Response**: We keep the description of a possible extension to 3D in Discussion but add 2D in the title and stress 2D in Introduction. See also the response to Major concern 1.

4. I am confused about the subgrid scheme, i.e., Eq (29). According to Line 267-268, the GL element is referred to the element that is partially grounded and partially floating. If it is true, then the Nitsche step function in Eq (29) should be 0, according to Line 301-302, "it is only imposed on the element which is fully on the ground" (I interpret it as "fully grounded"), and the lower panels in Fig. 2 and 3. Then, the right hand side of Eq. (29) is

$$\int_{\varepsilon_i} \mathcal{H}_{\beta} \beta \mathbf{u} \cdot \mathbf{v} + p_w \mathbf{n} \cdot \mathbf{v} \mathrm{d}s$$

This is different from Eq. (15) where the water pressure term is integrated partially from  $x_{GL}$  to  $x_i$ , instead of over the whole GL element. Also, I still don't follow where the friction step function of 1 / 2 is from. In the previous first round of the review, there was already a question about this, as it looks like a smoothing function than a partial integration using JUST the integration points in the grounded portion of the GL element. I don't get useful answers from the authors' response to this point.

**Response:** Corrections are made in (15) and (30).  $B_{\mathcal{N}} + B_{\Gamma}$ , F and  $F_{\Gamma}$  are separated. There are actually two levels of smoothing: the high order integration scheme in the subgrid model smooths the changes of the boundary conditions jumping from one node to another (between Eq. (5) and (6)); the 1/2 coefficient smooths the jump at the step function. The high order scheme acts on the friction law at the mesh size level with the step function  $\mathcal{H}_{\beta}$  and the 1/2 coefficient acts at the integration points level to smear out the rapid change of the step function.

**Minor points**

The abstract needs improvements. The abstract should cover the key points of the paper and the current version is still a bit ambiguous. My suggestion is it should at least cover the details of the improvements of the sub-grid method, compared to old model results.

**Response**: Two sentences have been added to Abstract.

The flow of the introduction section needs further care. For example, in the paragraph Line 28-34, the authors discussed a bit of different "model equations" (which looks a bit odd), and then in the paragraph Line 46-54, the authors discussed about different lower order models again. Would be nice to put them together so that the readers can easily follow the authors' logic here? Another example is that, in the paragraph Line 46-54, the authors discussed the sub-grid scheme and said "the purpose of a subgrid scheme is to avoid such fine meshes", and then in the next paragraph there is a similar sentence "Our subgrid scheme is aiming at improving the accuracy in GL simulations for a static mesh". It would be great if the authors make further organization for the introduction section.

**Response**: Introduction has been reorganized and partly rewritten.

I don't think the citation style is correct. For example, Line 14, "It is shown in (Kingslake et al., 2018)" should be "It is shown in Kingslake et al. (2018)". Similar mistakes should be corrected all over the whole paper.

**Response**: The corrections have been made.

• Line 12: remove "be able"

**Response**: The correction has been made.

• Line 13: change "sea" to "ocean"

**Response**: The correction has been made.

• Line 15: "km" - > "kilometers"

**Response**: The correction has been made.

• Line 18: The ice flow is dominated by vertical shear only when the basal friction is large.

**Response**: This is added now.

- Line 20: Any references for "gradual change of the stress field"? **Response**: We refer to Schoof 2011.
- Line 23: "interaction" -> "coupling"

**Response**: The correction has been made.

• Line 28: this sentence is unclear to me. I guess the authors try to say that different ice sheet models can generate different GL locations, which also depends on basal friction and some other numerical parameters. But it reads awkward.

**Response**: We have separated the model and numerical method in this statement. The sentence is rewritten.

- Line 31-32: "ice equations such as FS and SSA" is right. We can't say "equations such as Full Stokes and Shallow Shelf Approximation".
   **Response**: We have rewritten the sentence.
- Line 51-52: Need details of the subgrid modeling in Cornford et al., 2016 **Response**: Details have been included.
- Line 53-54: I suggest to remove the sentence "The purpose of ..." **Response**: It has been removed.
- Line 56: the sentence "Since the GL moves" duplicates with a similar sentence above

**Response**: The correction has been made.

- Line 77: I would suggest use "2D ice domain" to replace "2D vertical ice" **Response**: In the first round of referee reports, one referee suggested the phrase '2D vertical ice' and we keep it like that.
- Line 86: in Eq (1), g is a vector, not its z component. **Response**: The correction has been made.
- Line 87: Change the sentence "The rate factor..." to "The viscosity (η) is a function of the rate factor A(T). T is the ice temperature." **Response**: The correction has been made.
- Line 91: "vector t, see Fig. 1" -> "vector t (see Fig. 1)"
   Response: The correction has been made.
- Line 91: "In the 2D vertical case" -> "For the 2D ice domain", and similarly in the following

**Response**: See the response to line 77 above.

- Line 106: GL is the boundary (xGL, yGL) between ... **Response**: As defined in the text and shown in Fig. 1, the coordinate system in 2D is the x z plane. So, this should be (xGL, zGL).
- Line 108: "and g is ..." is a repeat of the description of g in Eq (1).
  Response: Changed. g is the vertical component of g.
- Line 128: "as observed also" -> "as also observed"
   Response: The correction has been made.
- Line 136: On the grounded ice domain, we have ... **Response**: The correction has been made.

- Line 152, switch the order of references according to the year. **Response**: The correction has been made.
- Line 261-262: "by" > "as"

**Response**: The correction has been made.

• Line 264-265: at the node xi

**Response**: The correction has been made.

• Figure 2: in the lower panel, why is the beta 1 / 2 at  $[x_{GL}, x_i]$  where the ice is floating?

**Response**: We have explanation in the paragraph preceding Eq (30) and in the response to Major concern 4.

• Line 292-294: The sentence "Moreover, this correction ... as discussed in Sect 3.3" is hard to understand. Can the authors provide more details/explanations? From my understanding, if we don't use a sub-grid scheme, wouldn't it be a slower advance?

**Response**: Sect 4 with the definitions of  $\chi$  and  $\tilde{\chi}$  is rewritten with better explanations.

• Line 298: "slip boundary" - > "grounded boundary"

**Response**: The 'slip boundary condition' refers to the boundary conditions of  $\Gamma_{bq}$  as defined in Eq. (5) which . We keep it as it is.

• Line 302: add comma after "On the contrary"

**Response**: The correction has been made.

There were no more comments after line 302 in the referee report we received.

[revised manuscript text omitted]
} \xrightarrow{n+1\ell+1} = z_{c} \xrightarrow{n\ell} + \Delta t (a_{c} \xrightarrow{n\ell} - u_{c} \frac{\partial z_{c}^{n+1}}{\partial x} \ell \frac{\partial z_{c}^{\ell+1}}{\partial x} + w_{c} \xrightarrow{n\ell}).$$

$$(20)$$

The spatial derivative of  $z_c$  is approximated by FEM as described above. A system of linear equations is solved at  $t^{n+1}$ 245 for  $z_c^{n+1}t^{\ell+1}$  for  $z_c^{\ell+1}$ . This time discretization and its properties are discussed in (Cheng et al., 2017) and summarized as Cheng et al. (2017) and summarized in Algorithm 2.

**Algorithm 2 Time scheme of the GL migration problem**

Start from an initial geometry  $\Omega^0$  defined by  $z_b^0, z_s^0$ . for  $\ell = 0$  to  $T/\Delta t - 1$  do Solve the FS equations on  $\Omega^\ell$  with Algorithm 1, to get the solution  $\mathbf{u}^\ell$ . Solve for  $z_b^{\ell+1}$  and  $z_s^{\ell+1}$  with  $\mathbf{u}^\ell$  by the semi-implicit Euler method. Use  $z_b^{\ell+1}$  and  $z_s^{\ell+1}$  to update  $\Omega^{\ell+1}$ .

The relation between  $u_{\mathbf{n}}$  and  $u_{\mathbf{t}}$  at  $\Gamma_{bf}$  and  $\mathbf{u}_{b} = \mathbf{u}(x, z_{b}(x))$  is

end for

A numerical stability problem in  $z_b$  is encountered in the boundary condition at  $\Gamma_{bf}$  when the FS equations are solved in (Durand et al., 2009a). It is resolved by expressing  $z_b$  in  $p_w$  at  $\Gamma_{bf}$  with a damping term. An alternative interpretation of the idea in (Durand et al., 2009a) Durand et al. (2009a) and an explanation follow below.

250

$$\mathbf{u}_{b} = \begin{pmatrix} u_{b} \\ w_{b} \end{pmatrix} = \begin{pmatrix} z_{bx} \\ -1 \end{pmatrix} \frac{u_{\mathbf{n}}}{\sqrt{1+z_{bx}^{2}}} + \begin{pmatrix} 1 \\ z_{bx} \end{pmatrix} \frac{u_{\mathbf{t}}}{\sqrt{1+z_{bx}^{2}}},\tag{21}$$

where  $z_{bx}$  denotes  $\partial z_b / \partial x$ . Inserting  $u_b$  and  $w_b$  from Eq. (21) into Eq. (8) yields

$$\frac{\partial z_b}{\partial t} = a_b - u_{\mathbf{n}} \sqrt{1 + z_{bx2}^2}.$$
(22)

Instead of discretizing Eq. (22) explicitly at  $t^{n+1}$  with  $u_n^n$  to determine  $p_w^{n+1}t^{\ell+1}$  with  $u_n^\ell$  to determine  $p_{w}^{\ell+1}$ , the base coordinate is updated implicitly

$$z_{b} \underbrace{\overset{n+1\ell+1}{\longrightarrow}}_{\sim} = z_{b} \underbrace{\overset{n\ell}{\longrightarrow}}_{\sim} + \Delta t \left( a_{b} \underbrace{\overset{n+1\ell+1}{\longrightarrow}}_{\sim} - u_{\mathbf{n}} \underbrace{\overset{n+1}{\longrightarrow}}_{\sim} \frac{1 + (z_{bx}^{n+1})^{2} \ell}{1 + (z_{bx}^{n+1})^{2}} \right)$$
(23)

[revised manuscript text omitted]

interpol

---

## Author Response (AR3)

**Response to the editor**

April 1, 2020

"Your revised manuscript is greatly improved, and I find that you have made satisfactory responses to the latest comments of the reviewers. I do have some minor concerns regarding readability, and the error analysis that I discuss below. However, pending these modifications and additions, I don't see too much barrier to publishing this paper, pending my final review.

The only substantive concern I have regards the error ana lysis, which one of the reviewers also brought up. I see that you have added a high resolution run, but the main way in which you address the error is by pointing out that you get similar results to previous papers with much lower resolution. While this is certainly impressive, it does not constitute an error analysis per se. For one, it does not preclude the possibility that you get these results "by accident", and also does not explain why your results don't appear to converge monotonically (i.e. there's some oscillation in the solution). It may be that you have converged already at these resolutions, and perhaps doing some coarser runs would show the convergence. The main point here is that I would like to see more discussion of the error and how it is changes with resolution. "

**Response**: No, we don't have any theoretical error analysis and are not aware of such an analysis elsewhere. The discontinuity at the boundary somehow affects the solution in the interior close to the GL. How the error behaves depends on the smoothness of the analytical solution and the finite element space and, in our case, on the subgrid treatment. What we have is an experimental investigation when the mesh size $\Delta x$ varies in Figs 7 and 8. The position of the GL is fairly constant for $\Delta x = 0.5, 1, 2, 4$ km in Fig 6. The solution in the ice in the neighborhood of the GL for the different mesh sizes depends regularly on the resolution in Figs 7 and 8.

"The rest of my suggested edits are primarily cosmetic and primarily to enhance readability. To cut down on the turn-around time for my decision, I have made these edits directly on the PDF, which is attached here. If you have any questions or do not understand these, do not hesitate to e-mail me. "

**Response**: We have responded to all your comments and suggestions. The details of our revision are found below.

**1 Changes**

- Page 1: All done.

- Page 2: All done.

- Page 3:

  line 72: A sentence is added.

  All done for the other comments, list is merged to the text.

- Page 4: All done

- Page 5

  line 112: We prefer to call $\beta$ a coefficient and reserve friction function for $\beta u_t$. In general, $\beta$ is not constant.

  line 116: changed.

- Page 6: All done

- Page 7: All done

- Page 8:

  line 177: Standard MISMIP setting, a ref to Durand 2019a is added.

  line 184: changed.

  line 193: The term is described in a better way.

- Page 12

  Fig 2: colors of lines are added, a ref to fig.2 is also added in fig.3 for the colors.

- Page 14

  line 318: A reference to Fig 3 is added.

- Page 15

  line 335: Ideally we would like to have a switch between 0 and 1, but this discontinuity will make it hard for the nonlinear iterations to converge. This is because nature of the finite element method: the high order integration scheme over a function (a Heaviside function in this case) can be considered as to approximate the function with a high order polynomial. Apparently, high order polynomials do not work well for the discontinuity in the Heaviside function. That is the reason of this additional smoothing.

  line 348: Three citations are added.

- Page 18

  Fig 6: We have new Figs 7 and 8 with convergence plots for mesh sizes $\Delta x = 0.5, 1, 2, 4$ km. There it appears as if the error behaves as $O(\Delta x)$ close to the GL. The GL position itself is less sensitive to $\Delta x$ in Fig 6. We comment on these results in Results and with one sentence in Conclusions.

  line 391: added.

- Page 20

  line 422: A few words are added. Dynamic adaptivity is less expensive than having a fine static mesh everywhere but there are some nontrivial issues to implement: solution of an adjoint problem, storing a full time dependent forward solution for the adjoint problem (could be 3D + time), redistribution of the mesh by splitting and collapsing mesh elements. This can be done and has been done but requires some effort in a large production quality code.

  line 424: One new sentence is added.

- Page 21: all done.

[revised manuscript text omitted]